# Time preferences are reliable across time-horizons and verbal versus experiential tasks

**Evgeniya Lukinova[1,2], Yuyue Wang[1,2], Steven F Lehrer[1,2,3,4], Jeffrey C Erlich[1,2,5]\***

[1]NYU-ECNU Institute of Brain and Cognitive Science at NYU Shanghai, Shanghai, China; [2]NYU Shanghai, Shanghai, China; [3]School of Policy Studies and Department of Economics, Queen's University, Kingston, Canada; [4]The National Bureau of Economic Research, Cambridge, United States; [5]Shanghai Key Laboratory of Brain Functional Genomics (Ministry of Education), East China Normal University, Shanghai, China

**\*For correspondence:**
jerlich@nyu.edu

**Competing interests:** The authors declare that no competing interests exist.

**Abstract** Individual differences in delay-discounting correlate with important real world outcomes, for example education, income, drug use, and criminality. As such, delay-discounting has been extensively studied by economists, psychologists and neuroscientists to reveal its behavioral and biological mechanisms in both human and non-human animal models. However, two major methodological differences hinder comparing results across species. Human studies present long time-horizon options verbally, whereas animal studies employ experiential cues and short delays. To bridge these divides, we developed a novel language-free experiential task inspired by animal decision-making studies. We found that the ranks of subjects' time-preferences were reliable across both verbal/experiential and second/day differences. Yet, discount factors scaled dramatically across the tasks, indicating a strong effect of temporal context. Taken together, this indicates that individuals have a stable, but context-dependent, time-preference that can be reliably assessed using different methods, providing a foundation to bridge studies of time-preferences across species.

**Editorial note:** This article has been through an editorial process in which the authors decide how to respond to the issues raised during peer review. The Reviewing Editor's assessment is that all the issues have been addressed (see decision letter).

DOI: https://doi.org/10.7554/eLife.39656.001

## Introduction

Intertemporal choices involve a trade-off between a larger outcome received later and a smaller outcome received sooner. Many individual decisions have this temporal structure, such as whether to purchase a cheaper refrigerator, but forgo the ongoing energy savings. Since research has found that intertemporal preferences are predictive of a wide variety of important life outcomes, ranging from SAT scores, graduating from college, and income to anti-social behaviors, for example gambling or drug abuse (*Frederick et al., 2002*; *Madden and Bickel, 2010*; *Casey et al., 2011*; *Golsteyn et al., 2014*; *Åkerlund et al., 2016*), they are frequently studied in both humans and animals across multiple disciplines, including marketing, economics, psychology, psychiatry, and neuroscience.

A potential obstacle to understanding the biological basis of intertemporal decision-making is that human studies differ from non-human animal studies in two important ways: long versus short time-horizons and choices that are made based on verbal versus non-verbal (i.e. 'experiential') stimuli. In animal studies, subjects experience the delay between their choice and the reward (sometimes

cued with a ramping sound or a diminishing visual stimulus) before they can proceed to the next trial (*Cai et al., 2011*; *Blanchard et al., 2013*; *Tedford et al., 2015*). Generally, there is nothing for the subject to do during this waiting period. In human studies, subjects usually make a series of choices (either via computer or a survey, often hypothetical) between smaller sooner and larger offers delayed by months or years (*McClure et al., 2004*; *Andersen et al., 2014*). (We are aware of only a handful of studies that have used delays of minutes (*McClure et al., 2007*) or seconds (*Lane et al., 2003*; *Gregorios-Pippas et al., 2009*; *Prevost et al., 2010*; *Tanaka et al., 2014*; *Fung et al., 2017*)). During the delay (e.g. if the payout is in 6 months) the human subjects go about their lives, likely forgetting about the delayed payment, just as individuals do not actively think about their retirement savings account each moment until their retirement.

Animal studies of delay-discounting take several forms (*Dalley et al., 2011*; *Redish et al., 2008*; *Cai et al., 2011*; *Wikenheiser et al., 2013*), but all require experiential learning that some non-verbal cue is associated with waiting. Subjects experience the cues, delays and rewards, and slowly build an internal map from the cues to the delays and magnitudes. Subjects may only have implicit knowledge of the map, which likely engage distinct neural substrates to the explicit processes engaged by humans when considering a verbal offer (*Reber et al., 2003*; *Poldrack et al., 2001*).

Whether animal studies can inform human studies depends on answers to the following questions. Do decisions that involve actively waiting for seconds invoke the same cognitive and neural processes as decisions requiring passively waiting for months? Do decisions made based on experience and perceptual decisions invoke the same cognitive and neural processes as decisions that are made based on explicitly written information?

The animal neuroscience literature on delay-discounting mostly accepts as a given that the behavior of animals will give insight into the biological basis for human impulsivity (*Fineberg et al., 2010*; *Huang et al., 2015*; *Schoenbaum et al., 2009*; *Robison and Nestler, 2011*) and rarely (*Blanchard et al., 2013*; *Rosati et al., 2007*; *Vanderveldt et al., 2016*) addresses the methodological gaps considered here. This view is not unfounded. Neural recordings from animals (*Cai et al., 2011*) and brain imaging studies in humans (*McClure et al., 2004*; *Kable and Glimcher, 2007*) both find that the prefrontal cortex and basal ganglia are involved in delay-discounting decisions, suggesting common neural mechanisms. Animal models of attention-deficit hyperactive disorder (ADHD) have reasonable construct validity: drugs that shift animal behavior in delay-discounting tasks can also improve the symptoms of ADHD in humans (*Paterson et al., 2012*; *Fineberg et al., 2010*). Thus, most neuroscientists would likely predict that our experiments would find high within-subject reliability across both time-horizons and verbal/experiential dimensions.

Reading the literature from economics, a different picture emerges. Traditional economic models (e.g. *Samuelson, 1947*) posit that agents make consistent intertemporal decisions, thereby implying a constant discount rate regardless of context. In contrast, growing evidence from behavioral economics provides support for the view that discounting over a given time delay changes with the time-horizon (*Berns et al., 2007*; *Andreoni et al., 2015*). Among human studies comparing short and long time-horizons only a few are within subject and incentivized, leaving this matter unresolved (*Paglieri, 2013*; *Johnson et al., 2015*; *Vanderveldt et al., 2016*; *Horan et al., 2017*). Yet, there remains debate in the empirical economics literature about how well discounting measures elicited in human studies truly reflect the rates of time-preference used in real-world decisions since measured discount rates have been found to vary by the type of task (hypothetical, potentially real, and real), stakes being compared, age of participants and across different domains (*Chapman and Elstein, 1995*). Thus, most economists surveying the empirical evidence would be surprised if a design that varied both type of tasks and horizons would generate results with high within-subject reliability.

Here, we have addressed these questions by measuring the discount factors of human subjects in three ways. First, we used a novel language-free task involving experiential learning with short delays. To our knowledge, this is the first time the time-preferences of human subjects have been measured in this way (*Vanderveldt et al., 2016*). Then, we measured discount factors more traditionally, with verbal offers over both short and long delays. This design allowed us to test whether, for each subject, a single process is used for intertemporal choice regardless of time-horizon or verbal vs. experiential stimuli, or whether the choices in different tasks could be better explained by distinct underlying mechanisms.

# Results

In our main experiment, 63 undergraduate students from NYU Shanghai participated in five experimental sessions. In each session, subjects completed a series of intertemporal choices. Across sessions, at least 160 trials in each task were conducted after learning (Materials and methods, *Figure 1—figure supplement 1*). In each trial, irrespective of the task, subjects made a decision between the sooner (blue circle) and the later (yellow circle) options. In the non-verbal task (*Figure 1A*), the parameters of the later option were mapped to an amplitude modulated pure tone. The reward magnitude was mapped to frequency of the tone (larger reward $\propto$ higher frequency). The delay was mapped to amplitude modulation rate (longer delay $\propto$ slower modulation). Across trials, the delay and the magnitude of the sooner option were fixed (4 coins, immediately), later options were drawn from all possible pairs of 5 magnitudes and delays (25 different offers, Materials and methods). For the short delay tasks, when subjects chose the later option, a clock appeared on the screen, and only when the clock image disappeared, could they collect their reward by clicking in the reward port. After clicking the reward port, the chosen number of coins appeared at the reward port and then a 'dropping coins' sound was played as the coins were added to a stack of coins on the right side of the screen that accumulated over the session. This stack gave subjects a visual indication of the total amount of rewards they had earned in the session. At the end of the session, the coins were converted to RMB as payment to the subject.

In the verbal tasks, the verbal description of the offers appeared within the blue and yellow circles in place of the amplitude modulated sound (*Figure 1B*). In the verbal long delay task, after each choice, subjects were given feedback confirming their choice (e.g. "Your choice: 8 coins in 30 days") and then proceeded to the next trial. Unlike the short tasks, there was no sound of dropping coins nor visual display of coins. At the end of the session, a single long-verbal trial was selected randomly to determine the payment (e.g. a subject was notified that "Trial 10 from session one was randomly chosen to pay you. Your choice in that trial was 8 coins in 30 days"). If the selected trial corresponded to a subject having chosen the later option, she received her reward via an electronic transfer after the delay (e.g. in 30 days).

## Subjects' time-preferences are reliable across both verbal/experiential and second/day differences

Subjects' impulsivity was estimated by fitting their choices with a Bayesian hierarchical model (BHM) of hyperbolic discounting with decision noise. The model had six population level parameters (log discount factor, $\log(k)$, and decision noise, $\tau$, for each of the three tasks, also known as fixed effects) and four parameters per subject: $\log(k_{NV})$, $\log(k_{SV})$, $\log(k_{LV})$ and $\tau$. We used this model to fit 32,707 choices across 63 subjects in the three tasks. We use the natural log of $k$, $\log(k)$, and not $k$ as a model parameter because we found that $k$ is approximately log-normally distributed over our subjects (as in *Sanchez-Roige et al., 2018*). The subject level effects are drawn from a normal distribution with mean zero. In other words, the subject level effects reflect the difference of each subject relative to the mean across subjects. As such, the actual discount factor for the $n^{th}$ subject in the SV task, $k_{n,SV} = e^{\log(\hat{k}_{SV}) + \log(\dot{k}_{n,SV})} = \hat{k}_{SV} \cdot \dot{k}_{n,SV}$, where $\log(\hat{k}_{SV})$ represents the population level log discount factor for SV and $\log(\dot{k}_{n,SV})$ represents the subjects level effect for subject $n$ in SV. For the sake of brevity, we refer to 'log discount factor' as 'discount factor' throughout the text.

The population level parameters reflect the mean over all subjects. For example, if the mean discount factor across subjects was equal in all tasks, then the population level discount factor parameters would also be equal. If all subjects were exactly twice as impulsive in short vs. non-verbal tasks, then that change would be reflected in the population level discount factor ($k_{SV} = 2 \cdot k_{NV} \rightarrow \log(k_{SV}) = \log(k_{NV}) + \log(2)$), and the subject level parameters would be the same across tasks. If, on the other hand, impulsive subjects (relative to the mean) became more impulsive, and patient subjects became more patient, that would result in clear changes to subject level parameters, with relatively little change in the population level parameters (assuming the same scaling factor for impulsive and patient subjects).

Subjects' choices were well-fit by the model (*Figure 2*, *Figure 2—figure supplement 1*, *Supplementary file 1*). Since we did not *ex ante* have a strong hypothesis about how the subjects' impulsivity measures in one task would translate across tasks, we fit subjects' choices in the units of the task (i.e. seconds or days), examined ranks of impulsivity at first and found significant correlations

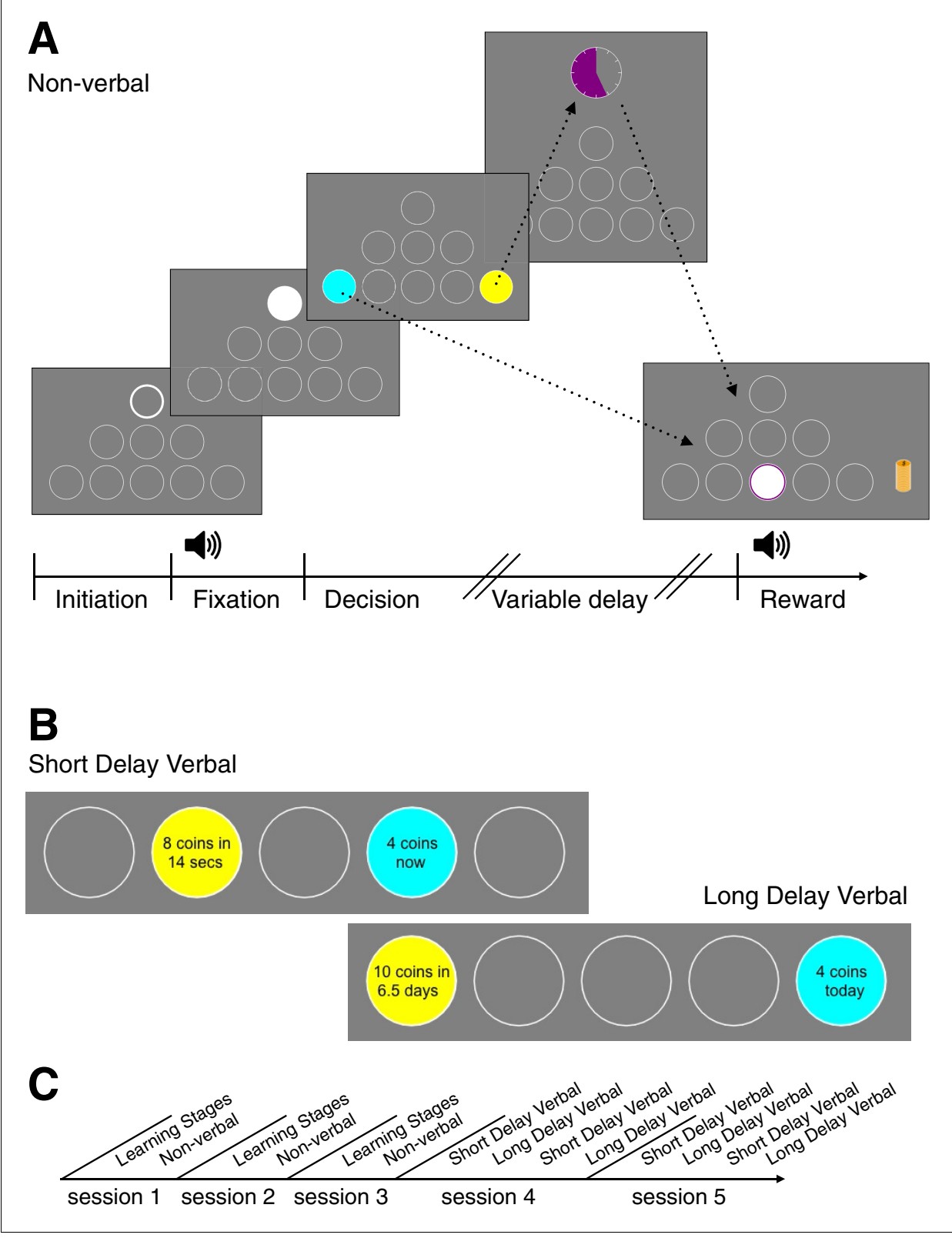

**Figure 1.** Behavioral Tasks. (**A**) A novel language-free intertemporal choice task. This is an example sequence of screens that subjects viewed in one trial of the non-verbal task. First, the subject initiates the trial by pressing on the white-bordered circle. During fixation, the subject must keep the cursor inside the white circle. The subject hears an amplitude modulated pure tone (the tone frequency is mapped to reward magnitude and the modulation rate is mapped to the delay of the later option). The subject next makes a decision between the sooner (blue circle) and later (yellow circle) options. If
*Figure 1 continued on next page*

*Figure 1 continued*

the later option is chosen, the subject waits until the delay time finishes, which is indicated by the colored portion of the clock image. Finally, the subject clicks in the middle bottom circle ('reward port') to retrieve their reward. The reward is presented as a stack of coins of a specific size and a coin drop sound accompanies the presentation. (B) Stimuli examples in the verbal experiment during decision stage (the bottom row of circles is cropped). (C) Timeline of experimental sessions. Note: The order of short and long delay verbal tasks for sessions 4 and 5 was counter-balanced across subjects.

DOI: https://doi.org/10.7554/eLife.39656.002

The following figure supplement is available for figure 1:

**Figure supplement 1.** Learning stages example performance.

DOI: https://doi.org/10.7554/eLife.39656.003

across experimental tasks (*Table 1*). In other words, the most impulsive subject in one task was likely to be the most impulsive subject in another task. This result is robust to different functional forms of discounting (e.g. hyperbolic vs. exponential) and estimation (e.g. Bayesian hierarchical models vs. fitting subjects individually using maximum likelihood estimation vs. model-free) methods (*Figure 2—figure supplement 1*, *Figure 2—figure supplement 2*, *Figure 2—figure supplement 3*). For example, if we ranked the subjects by the fraction of trials in which they chose the later option in each task, we obtained a similar result (Spearman $r$: SV vs. NV $r = 0.71$; SV vs. LV $r = 0.49$; NV vs. LV $r = 0.30$, all $p < 0.05$). The correlations of discount factors across tasks extended to Pearson correlation of $\log(k)$ (*Figure 3*, *Table 1*). This indicates that subjects' preferences are reliable across the verbal/experiential gap and time-horizons.

Having addressed our initially planned analysis, we continued with analyses to further understand the subjects' choices within and across the tasks. Consistent with existing research, we found that time-preferences were stable in the same task within subjects between the first half of each reward block and the second half of the block within sessions (time-preferences are measured as % of yellow choices, Wilcoxon signed-rank test, $p = 0.35$; Pearson $r = 0.81$, $p < 10^{-9}$) and also across experimental sessions that take place every two weeks: % of yellow choices between NV sessions (Wilcoxon signed-rank test, $p = 0.47$; Pearson $r = 0.7$, $p < 10^{-9}$), between SV sessions (Wilcoxon signed-rank test, $p = 0.66$; Pearson $r = 0.82$, $p < 10^{-9}$) and a slight difference between LV sessions (Wilcoxon signed-rank test, $p < 0.1$; Pearson $r = 0.66$, $p < 10^{-9}$) (*Meier and Sprenger, 2015*; *Augenblick et al., 2015*). In our verbal experimental sessions, the short and long tasks were alternated and the order was counter-balanced across subjects. We did not find any order effects in either main (bootstrapped mean test, SV-LV-SV-LV vs. LV-SV-LV-SV for SV and LV $\log(k)$, all $p > 0.4$) or control experiments (NC, bootstrapped mean test, SV-LV-SV-LV vs. LV-SV-LV-SV for SV and LV $\log(k)$, all $p > 0.6$; DW, bootstrapped mean test, DV-WV-DV-WV vs. WV-DV-WV-DV for DV and WV $\log(k)$, all $p > 0.2$).

In addition to the reliability of subjects' choices, other aspects of their behavior were also consistent. We examined the total time it took subjects to finish each session. This time includes waiting time (i.e. the chosen delays in the short task) and also non-waiting time (i.e. intertrial intervals and subject reaction times). The total time taken did not change significantly across sessions (bootstrapped mean tests: between NV session 2 and 3, $p = 0.55$; between verbal sessions 1 and 2, $p = 0.08$). By definition, the waiting time is correlated with $\log(k)$. But we also found that for the short sessions non-waiting time (and total-time) were correlated with $\log(k)$ and also the fraction of total reward earned (relative to a subject that always picked the larger offer regardless of time; *Figure 3—figure supplement 1*). This suggests that impulsive subjects not only express their impatience in their choices of a sooner option, but also make their choices faster.

In our experimental design, the SV task has shared features with both the NV and LV task. First, the SV shares time-horizon with the NV task. Second, the SV and LV are both verbal and were undertaken at the same time, always following NV task. The NV and LV tasks differ in both time-horizon and verbal/non-verbal. The central feature that is shared between all tasks is delay-discounting. To test whether the correlation between NV and LV might be accounted for by their shared correlation with the SV task, we performed linear regressions of the discount factors in each task as a function of the other tasks (e.g. $\log(k_{NV}) = \beta_{SV} \log(k_{SV}) + \beta_{LV} \log(k_{LV}) + \beta_0 + \epsilon$ ). For *NV* the two predictors explained 63% of the variance ($F(60,2) = 50.63$, $p < 10^{-9}$). It was found that $\log(k_{SV})$ significantly predicted $\log(k_{NV})$ ($\beta_{SV} = 1.28 \pm 0.15$, $p < 10^{-9}$) but $\log(k_{LV})$ did not ($\beta_{LV} = -0.12 \pm 0.09$, $p = 0.18$). For *LV* we were able to predict 40% of the variance ($F(60,2) = 19.64$, $p < 10^{-6}$) and found that $\log(k_{SV})$

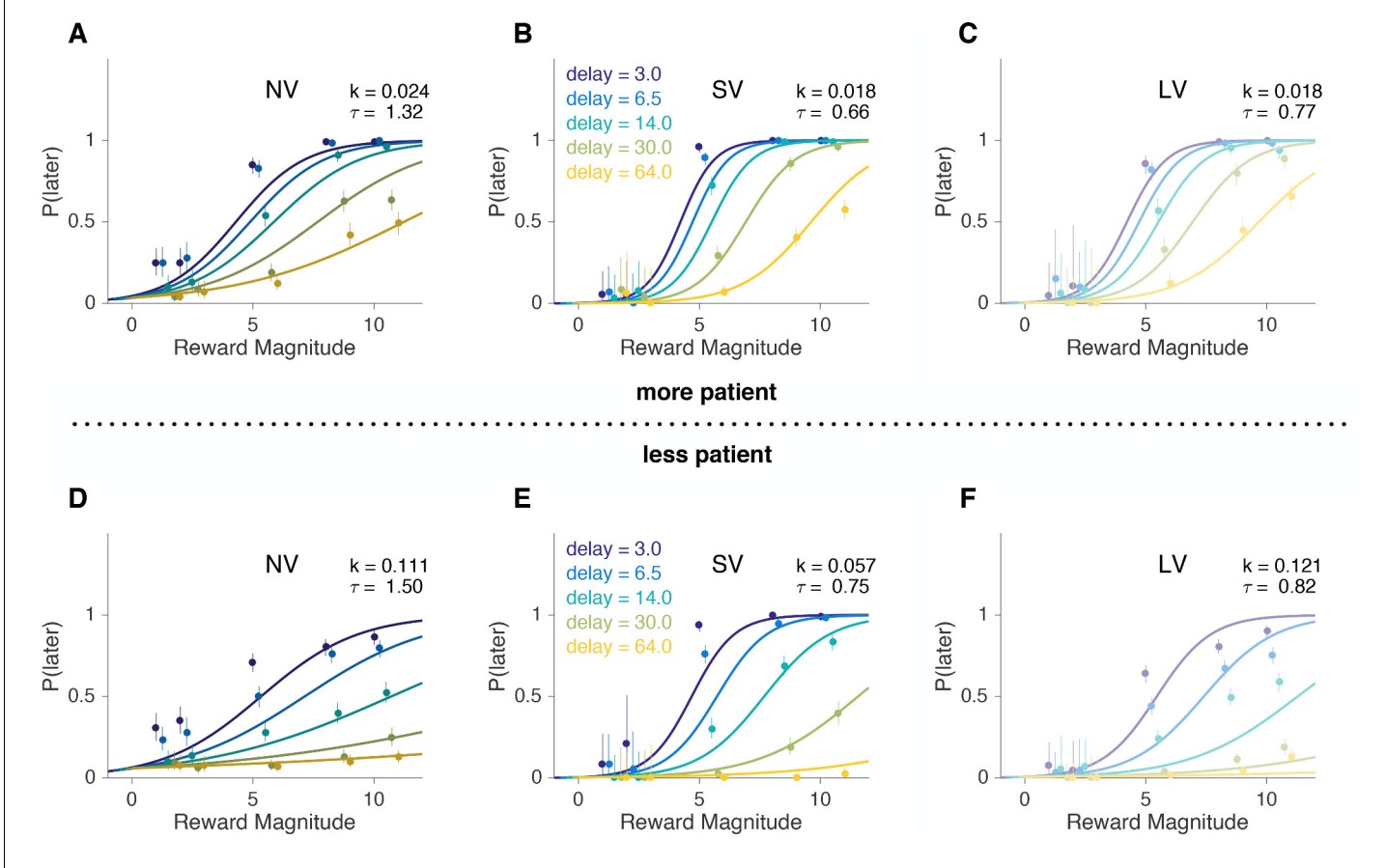

**Figure 2.** A 50% median split (±1 standard deviation) of the softmax-hyperbolic fits. (A–C) more patient and (D–F) less patient subjects. The values of $k$ and $\tau$ are the means within each group. Average psychometric curves obtained from the model fits (lines) versus actual data (circles with error bars) for NV, SV and LV tasks for each delay value, where the x-axis is the reward magnitude and the y-axis is the probability (or proportion for actual choices) of later choice. Error bars are binomial 95% confidence intervals. We excluded the error in the model for visualization. Note: The lines here are not a model fit to aggregate data, but rather reflect the mean model parameters for each group. As such, discrepancies between the model and data here are not diagnostic. See individual subject plots (*Supplementary file 1*) to visualize the quality of the model fits.
DOI: https://doi.org/10.7554/eLife.39656.004

The following figure supplements are available for figure 2:

**Figure supplement 1.** An example of the softmax-hyperbolic fit for one subject in Matlab and Stan.
DOI: https://doi.org/10.7554/eLife.39656.005
**Figure supplement 2.** BHM fits vs. Matlab fits.
DOI: https://doi.org/10.7554/eLife.39656.006
**Figure supplement 3.** Non-parametric out-of-sample prediction.
DOI: https://doi.org/10.7554/eLife.39656.007
**Figure supplement 4.** The role of parameters in hyperbolic utility model with softmax.
DOI: https://doi.org/10.7554/eLife.39656.008

significantly predicted $\log(k_{LV})$ ($\beta_{SV} = 1.26 \pm 0.26$, $p < 10^{-5}$) but $\log(k_{NV})$ did not ($\beta_{NV} = -0.24 \pm 0.18$, $p = 0.18$). For $SV$ the two predictors explained 72% of the variance ($F(60, 2) = 78.93$, $p < 10^{-9}$). Coefficients for both predictors were significant ($\beta_{NV} = 0.44 \pm 0.05$, $p < 10^{-9}$; $\beta_{LV} = 0.22 \pm 0.05$, $p < 10^{-5}$); where $\beta = mean \pm std.error$.

We further checked whether the correlations between discount factors in the three tasks may have arisen due to some undesirable features of our task design. For example, different subjects experienced the offers in different orders. Anchoring effects (*Tversky and Kahneman, 1974*; *Wilson et al., 1996*; *Furnham and Boo, 2011*) may have set a reference point in the early part of the experiment that guided choices throughout the rest. As such, we repeated the analyses

**Table 1.** Correlations of subjects' discount factors [95% CI].

Corrected rank correlations of subjects' discount factors were normalized using simulations to estimate the expected maximum correlation we could observe (*Figure 3—figure supplement 4*). The correlations between each task were significantly different from each other at $p < 0.05$ using various methods as in the R package 'cocor' (*Diedenhofen and Musch, 2015*).

| | Spearman Rank Correlation | Corrected Rank Correlation | Pearson Correlation |
|---|---|---|---|
| SV vs. NV | 0.76 [0.61, 0.85] | 0.77 [0.62, 0.87] | 0.79 [0.65, 0.88] |
| SV vs. LV | 0.54 [0.30, 0.73] | 0.57 [0.31, 0.77] | 0.61 [0.41, 0.76] |
| NV vs. LV | 0.36 [0.11, 0.57] | 0.39 [0.12, 0.62] | 0.40 [0.18, 0.60] |

all $p < 0.01$

DOI: https://doi.org/10.7554/eLife.39656.014

described in the previous paragraph, but we added six additional factors: the mean rewards and delays presented in the first block of the 2nd and 3rd non-verbal session and also the % of yellow choices made in those blocks. We reasoned that if anchoring effects were playing a role then subjects that were presented with longer delays, or smaller rewards early in the experiment should have correlations between these factors and $\log(k_{SV})$ or $\log(k_{LV})$. Likewise, if subjects were simply trying to be consistent with their early choices, then the '% yellow' in the early reward blocks would have an

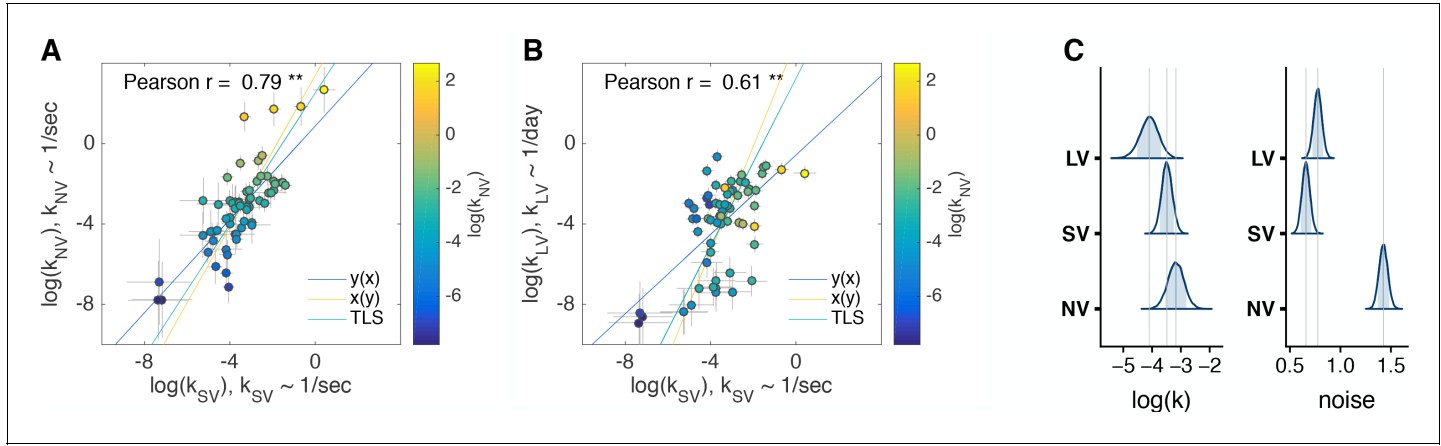

**Figure 3.** Comparison of discount factors across three tasks in the main experiment. (A, B) Each circle is one subject (N = 63). The logs of discount factors in SV task (x-axis) plotted against the logs of discount factors in NV (A) and LV (B) tasks (y-axis). The color of the circles and the colorbar identify the ranksdiscount factors in NV task. Pearson's $r$ is reported on the figure ($p < 0.01$ - '**'). The error bars are the SD of the estimated coefficients (posterior means). Three lines (*Huang et al., 2013*) represent the vertical $y(x)$, horizontal $x(y)$ and perpendicular (or total) least squares (TLS) regression lines. (C) Distribution of posterior parameter estimates of $\log(k)$ and decision noise $\tau$ from the model fit for the three tasks in the main experiment ($k_{NV} \sim 1/\sec$, $k_{SV} \sim 1/\sec$, $k_{LV} \sim 1/day$). The light blue shaded area marks the 80% interval of the posterior estimate. The outline of the distribution extends to the 99.99% interval. Thin grey lines are drawn through the mean of each distribution to ease comparison across tasks. Comparisons between tasks are reported in *Table 3*. Note, the units for $k_{SV}$ & $k_{NV}$ ($1/\sec$) would need to be scaled by $86400 secs/day \rightarrow \log(86400) = 11.37$ to be directly compared to $k_{LV}$.

DOI: https://doi.org/10.7554/eLife.39656.009

The following figure supplements are available for figure 3:

**Figure supplement 1.** Model-free analysis of short tasks.

DOI: https://doi.org/10.7554/eLife.39656.010

**Figure supplement 2.** Ruling out the anchoring effect.

DOI: https://doi.org/10.7554/eLife.39656.011

**Figure supplement 3.** Obtained subjective utilities and hyperbolic fits of individual subjects.

DOI: https://doi.org/10.7554/eLife.39656.012

**Figure supplement 4.** Simulation results: distributions of expected correlations of the discount factors ranks between tasks.

DOI: https://doi.org/10.7554/eLife.39656.013

important influence. We tested the contribution of each factor by dropping it from the model to create a reduced nested model and using a likelihood ratio test against the full model (*Figure 3—figure supplement 2*). We found no evidence for anchoring effects or that subjects were simply trying to be consistent with their early choices.

In order to test whether the verbal/non-verbal gap or the time-horizons gap accounted for more variation in discounting, we used a linear mixed-effects model where we estimated $\log(k)$ as a function of the two gaps (as fixed effects) with subject as a random effect, using the 'lme4' R package (*Bates et al., 2014*). We created two predictors: *days* was false in NV and SV tasks for offers in seconds and was true in the LV task for offers in days; *verbal* was true for the SV and LV tasks and false for the NV task. We found that time-horizon ($\beta_{days} = -0.52 \pm 0.24$, $p = 0.03$) but not verbal/non-verbal ($\beta_{verbal} = -0.32 \pm 0.24$, $p = 0.18$) contributed significantly to the variance in $\log(k)$. This result was further supported by comparing the two-factor model with reduced one-factor models (i.e. that only contained either time or verbal fixed effects). Dropping the *days* factor significantly decreased the likelihood, but dropping the *verbal* factor did not (*Table 2*).

We described above that subject's time-preferences were highly correlated across tasks. However, correlation is invariant to shifts or scales across tasks. Our hierarchical model allows us to directly estimate the posterior distributions of $\log(k)$ and $\tau$ (*Figure 3C*) and report posterior means and 95% credible intervals ($\log(k)$ NV = $-3.2$ [-3.77,−2.64], SV = $-3.49$ [$-3.86$, $-3.11$], LV = $-3.95$ [$-4.55$, $-3.34$]). Note, that $k_{NV}$ and $k_{SV}$ have units of Hz ($1/s$), but $k_{LV}$ has units of $1/day$. Thus, while the 95% credible intervals of the means of $\log(k)$ are overlapping for the three tasks when expressed in the units of each task, the mean $\log(k_{LV})$ is in fact shifted to $-14.86$ when $k_{LV}$ is expressed in units of 1/s. We further analyze and discuss this scaling subsequently, but first we compare $\log(k)$ in the units of each task, in consideration of subjects potentially ignoring the time units in their choices (*Furlong and Opfer, 2009*; *Cox and Kable, 2014*). We find that, on average, subjects were most patient in LV, then SV then NV *Table 3*). Note, that a shift of 1 log-unit is substantial. For example, a subject with $\log(k_{SV}) \approx -3$ would value 10 coins at half its value in just 20 s. But for $\log(k_{SV}) \approx -4$ the coins would lose half their value in 55 s (*Figure 3—figure supplement 3*).

In addition to the shift, we observed significant scaling of $\log(k)$ between SV and the other two tasks (*Table 3*, note: scaling is insensitive to the units of $k$, since $\log(C \cdot k) = \log(C) + \log(k)$). This is likely driven by subgroups that were exceptionally patient in the LV task (*Figure 3B*) or impulsive in the NV task (*Figure 3A*). We also observed a clear increase in the decision noise in the NV task, $\tau_{NV}$, compared to the other two tasks (*Figure 3C*), which is unsurprising given that in NV subjects have to make a perceptual decision (mapping the sound features to delay and magnitude) in addition to an economic decision. However, even in the verbal tasks subjects show stochasticity in choice. This is clearly evident for the longer delays (*Supplementary file 1*).

## Controlling for visuo-motor confounds

In the main experiment, we held the following features constant across three tasks: the visual display and the use of a mouse to perform the task. However, after observing the strong correlations between the tasks (*Figure 3*) we were concerned that the effects could have been driven by the superficial (i.e. visuo-motor) aspects of the tasks. In other words, the visual and response features of the SV and LV tasks may have reminded subjects of the NV task context and nudged them to use a similar strategy across tasks. While this may be interesting in its own right, it would limit the generality of our results. To address this, we ran a control experiment (n = 25 subjects) where the NV task was identical to the original NV task, but the SV and LV tasks were run in a more traditional way, with a text display and keypress response (control experiment 1, Materials and methods, *Figure 4—*

**Table 2.** Relative contributions of two gaps to variance in $\log(k)$ (two-factor model comparison with two reduced one-factor models).

| Dropped factor | $\Delta df$ | AIC | LR test $p$ |
|---|---|---|---|
| *none* | | 743.06 | |
| *verbal* | 1 | 742.88 | 0.18 |
| *days* | 1 | 745.99 | 0.03 |

DOI: https://doi.org/10.7554/eLife.39656.015

**Table 3.** Shift and scale of $\log(k)$ between tasks.

$k_{SV}, k_{NV} \sim (1/s)$. $k_{LV} \sim (1/day)$. The evidence ratio (Ev. Ratio) is the Bayes factor of a hypothesis vs. its alternative, for example $P(a>b)/P(a<b)$. '*'denotes $p<0.01$, one-sided test. Expressing $\log(k_{LV})$ in units of 1/s (for direct comparison with the other tasks) results in a negative shift in $\log(k_{LV})$ and even larger differences in means without changing the difference between standard deviations.

| Comparison | $log_2(Ev.\ Ratio)$ |
|---|---|
| **between means** | |
| $\mu\ \log(k_{SV}) \boldsymbol{>} \mu\ \log(k_{LV})$ | 6.79 * |
| $\mu\ \log(k_{NV}) \boldsymbol{>} \mu\ \log(k_{LV})$ | 7.92 * |
| $\mu\ \log(k_{NV}) \boldsymbol{>} \mu\ \log(k_{SV})$ | 4.16 |
| **between standard deviations** | |
| $\sigma\ \log(k_{LV}) \boldsymbol{>} \sigma\ \log(k_{SV})$ | 8.43 * |
| $\sigma\ \log(k_{NV}) \boldsymbol{>} \sigma\ \log(k_{LV})$ | 0.92 |
| $\sigma\ \log(k_{NV}) \boldsymbol{>} \sigma\ \log(k_{SV})$ | 11.48 * |

DOI: https://doi.org/10.7554/eLife.39656.016

*figure supplement 1*). We replicated the main findings of our original experiment for ranks of $\log(k)$ (*Figure 4*) and correlation between $\log(k)$ in SV and LV tasks (*Figure 4B*). To determine whether the correlations observed were within the range expected by chance (given the difference in sample size), we repeatedly (10,000 times) randomly sampled 25 of the original 63 subjects (from *Figure 3*) and computed the correlations between tasks. Pearson's $r = .42$ is lower than we would expect for NC (the 95% CI of the correlation between SV and NV in the main experiment assuming 25 subjects is [0.50 0.92]). This suggests that some of the correlation between SV and NV tasks in the main experiment may be driven by visuo-motor similarity in experimental designs. We did not find shifts

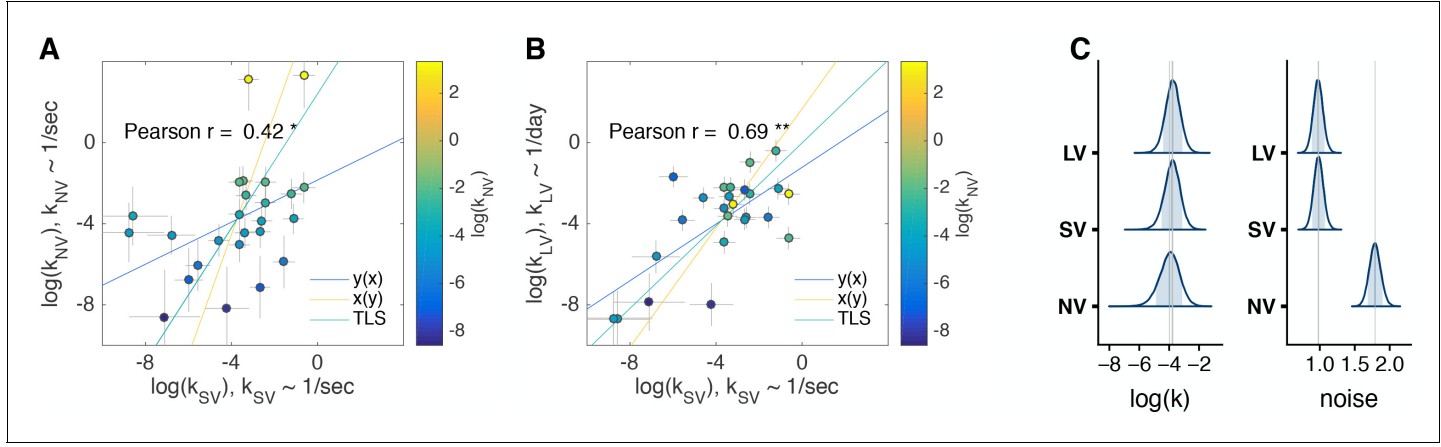

**Figure 4.** Comparison of discount factors across three tasks in control experiment 1. (A,B) Control experiment 1 (n = 25). The logs of discount factors in SV task (x-axis) plotted against the logs of discount factors in NV (**A**) and LV (**B**) tasks (y-axis). The color of the circles and the colorbar identify the discount factors in NV task. Each circle is one subject. Pearson's $r$ is reported on the figure ($p<0.01$ - '**', $p<0.05$ - '*'). Spearman $r$: SV vs. NV $r = 0.52$; SV vs. LV $r = 0.52$ (all $p<0.01$). The error bars are the SD of the estimated coefficients. Three lines represent the vertical $y(x)$, horizontal $x(y)$ and total least squares (TLS) regression lines. See individual subject plots (*Supplementary file 2*) to visualize the quality of the model fits. (**C**) Distribution of posterior parameter estimates of $\log(k)$ and decision noise $\tau$ from the model fit for the three tasks in control experiment 1 ($k_{NV} \sim 1/s$, $k_{SV} \sim 1/s$, $k_{LV} \sim 1/day$). The light blue shaded area marks the 80% interval of the posterior estimate. The outline of the distribution extends to the 99.99% interval. Thin grey lines are drawn through the mean of each distribution to ease comparison across tasks.

DOI: https://doi.org/10.7554/eLife.39656.017

The following figure supplement is available for figure 4:

**Figure supplement 1.** Control experiment 1 choice screen example.

DOI: https://doi.org/10.7554/eLife.39656.018

or scaling between the posterior distributions of $\log(k)$ across tasks in this control experiment (*Figure 4C*, mean [95% CI] NV = −3.98 [-5.44,–2.67], SV = −3.8 [−4.94, −2.75], LV = −3.76 [−4.79, −2.76]), but we found again that noise in NV was higher than in the other tasks.

## Controlling for differences in reward experience

We designed our non-verbal task so that with minimal changes we could use it in animals: rats and mice in particular. In rodent decision-making primary rewards are typically used (e.g. (*Carter and Redish, 2016*; *Wikenheiser et al., 2013*; *Erlich et al., 2015*)). In order to make the reward in the short tasks more like a primary reinforcer, we included visual and auditory cues at the time of the reward. This introduces a potential confound to one of our findings: that the correlation between the two short tasks is higher than the correlation between long and short tasks. It could be that inter-subject variability in the experience of the audio-visual cues could lower the correlation between the short and long tasks, but since it is shared between the two short tasks, those correlations would be artificially inflated. In order to address this, we refit our model with the following changes: we added a *reward scaling* parameter that multiplies with reward magnitude on each trial. This parameter has two levels (short/long) which can vary for each subject. This adds two population level parameters and $63 \times 2 = 126$ subject level parameters to the model. We compared the original and expanded model using 10-fold cross-validation ('kfold' function in the 'brms' R package). This process fits model parameters using 90% of the data and then produces a posterior predictive density for the left out 10% and repeats this 10 times (for each left out 10%). This procedure results in an expected log posterior density for the model (*Vehtari et al., 2017*), which is then multiplied by −2 to produce a K-fold Information Criteria (KfoldIC), as in other metrics like Akaike, Bayesian or deviance information criteria. The expanded model was substantially better than the original model ($\Delta KfoldIC \pm SE = 2207.40 \pm 81.06$, $r^2_{original} = 0.595 \pm 0.002$, $r^2_{expanded} = 0.640 \pm 0.002$). This is strong evidence that an important component of the intersubject variability in our task comes from differences in experience of the reward.

Having justified the additional parameters, we re-examined the correlations between $\log(k)$ in the three tasks in the expanded model. We found that the between task correlations were slightly larger but highly overlapping with the correlations in the original model (*Table 4*), thus supporting our findings about the relative reliability between tasks. The population $\log(k)$ and decision noise estimates also followed the same pattern as in the original model (compare *Figure 3C* with *Figure 5*). The $\log(k)$ estimates were shifted slightly higher (estimating subjects as more impulsive) with a corresponding increase in the experience of the rewards. That is, in both long and short tasks, the reward scaling was greater than one. Note, however, that reward scaling for long vs short tasks are not 5 orders of magnitude apart, so this cannot account for the massive scaling of discount factors between the long and short tasks.

**Table 4.** Correlations of subjects' log discount factors [95% CI] in the original model (taken from *Table 1*) and the expanded model which included differential reward scaling between the short and long tasks.

The correlations between each task were significantly different from each other at $p < 0.05$ for both the original and expanded models using various methods as in the R package 'cocor' (*Diedenhofen and Musch, 2015*).

| | Original model Pearson Correlation | Expanded model Pearson Correlation |
|---|---|---|
| SV vs. NV | 0.79 [0.65, 0.88] | 0.84 [0.75, 0.90] |
| SV vs. LV | 0.61 [0.41, 0.76] | 0.65 [0.48, 0.77] |
| NV vs. LV | 0.40 [0.18, 0.60] | 0.50 [0.29, 0.66] |

all correlations are significantly different from 0, $p < 0.01$

DOI: https://doi.org/10.7554/eLife.39656.020

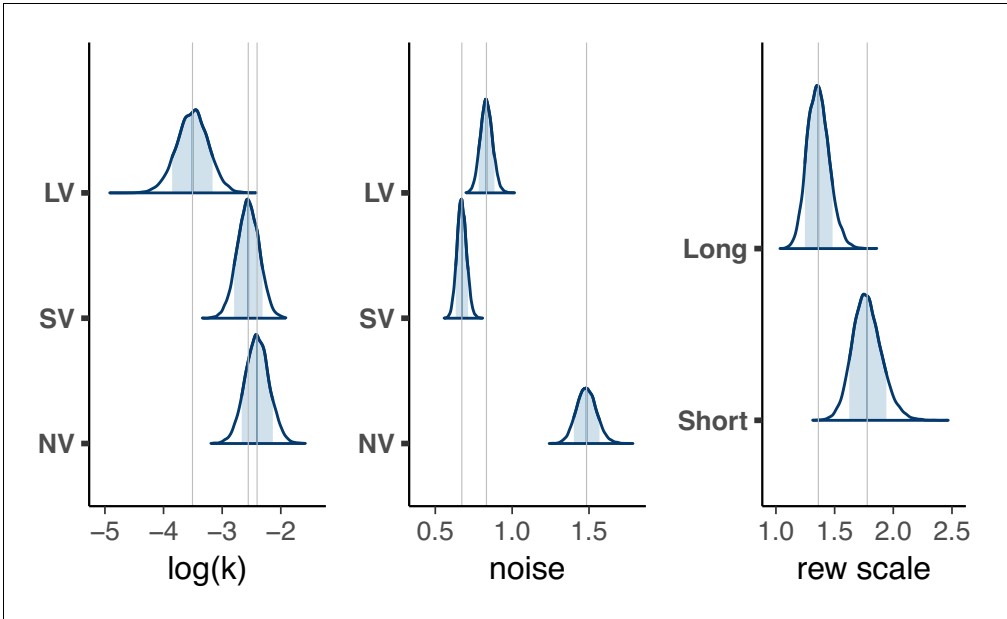

**Figure 5.** Distribution of population level posterior parameter estimates from the expanded model fit with reward scaling for the three tasks in the main experiment. The light blue shaded area marks the 80% interval of the posterior estimate. The outline of the distribution extends to the 99.99% interval. Thin grey lines are drawn through the mean of each distribution to ease comparison across tasks. Note, the units for $k_{SV}$ & $k_{NV}$ $(1/s)$ would need to be scaled by $86400 secs/day \rightarrow \log(86400) = 11.37$ to be directly compared to $k_{LV}$ $(1/day)$.

DOI: https://doi.org/10.7554/eLife.39656.019

## Strong effect of temporal context

As described above, we fit the discount factors for each task in the units of that task: $k_{SV}$ and $k_{NV}$ in units of seconds and $k_{LV}$ in units of days. Since there are 86,400 s in a day, classic economic theory would posit that we would find $\Delta \log(k) = 11.37$ between the long and short tasks to account for the difference in units. But, we found that the discount factors in the LV task, $k_{LV}$, were close to those in the other tasks (within $\approx 1$ log-unit) (*Figure 3C*). This finding implies that for a specific reward value, if a subject would decrease their subjective utility of that reward by 50% for a 10 s delay in the SV task, they would also decrease their subjective utility of that reward by 50% for a 10-*day* delay in the LV task. This seems incredible, particularly from a neoclassical economics perspective, but has been previously reported (*Navarick, 2004*; *Lane et al., 2003*). What could explain this scaling effect? In addition to the change in time units, reward units also changed between the short and long tasks. In our sessions, the exchange rates in NV and SV were 0.1 and 0.05 CNY per coin, respectively (since all coins are accumulated and subjects are paid the total profit), whereas in LV, subjects were paid on the basis of a single trial chosen at random using an exchange rate of 4 CNY for each coin. These exchange rates were set to, on average, equalize the possible total profit between short and long delays tasks. However, even accounting for both the magnitude effect (*Green et al., 1999*; *Green et al., 2004*) and unit conversion (calculations presented in Materials and methods) the discount rates are still scaled by 4 orders of magnitude from the short to the long time-horizon tasks (*Navarick, 2004*).

One possible explanation for this scaling is that subjects are simply ignoring the units and only focusing on the number. This would be consistent with an emerging body of evidence that numerical value, rather than conversion rate or units matter to human subjects (*Furlong and Opfer, 2009*; *Cox and Kable, 2014*). A second possible explanation is that subjects normalize the subjective delay of the offers based on context, just as they normalize subjective value based on current context and recent history (*Lau and Glimcher, 2005*; *Tymula and Glimcher, 2016*; *Louie et al., 2015*; *Khaw et al., 2017*). A third possibility is that in the short delay tasks (NV and SV), subjects experience the wait for the reward on each trial as quite costly, in comparison to the postponement of

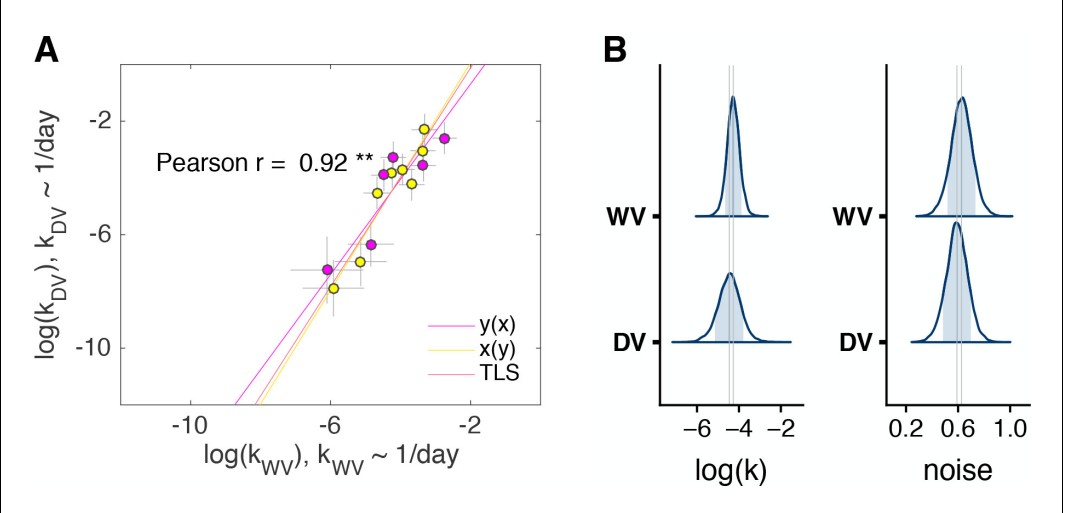

**Figure 6.** Control experiment 2. (**A**) The discount factors in WV task plotted against the discount factors in DV task, (n = 14, two out of 16 subjects who always chose the later option were excluded from the model). The color of the circles identifies the order of task appearance. Each circle is one subject. Pearson's $r$ is reported on the figure ($p < 0.01$ - '**'). The error bars are the SD of the estimated coefficients. Three lines represent the vertical $y(x)$, horizontal $x(y)$ and total least squares (TLS) regression lines. See individual subject plots (**Supplementary file 3**) to visualize the quality of the model fits. (**B**) Distribution of posterior parameter estimates of $\log(k)$ and decision noise $\tau$ from the model fit for the two tasks in control experiment 2 ($k_{DV} \sim 1/day$, $k_{WV} \sim 1/day$). The light blue shaded area marks the 80% interval of the posterior estimate. The outline of the distribution extends to the 99.99% interval. Thin grey lines are drawn through the mean of each distribution to ease comparison across tasks.
DOI: https://doi.org/10.7554/eLife.39656.021
The following figure supplement is available for figure 6:

**Figure supplement 1.** Subjective utilities as a function of the delay in days.
DOI: https://doi.org/10.7554/eLife.39656.022

reward in the LV task. This 'cost of waiting' may share some intersubject variability with delay-discounting but may effectively scale the discount factor in tasks with this feature (**Paglieri, 2013**).

To test the first hypothesis, that subjects ignore units of time, we ran a control experiment (n = 16 subjects) using two verbal discounting tasks (control experiment 2, Materials and methods). In one task, the offers were in days (DV). In the other, the offers were in weeks (WV). This way, we could directly test whether subjects would discount the same for 1 day as 1 week (i.e. ignore units) or 7 days as 1 week (i.e. convert units). For this experiment, we converted the delays from the weeks task into days (i.e. delay in days = 7× delay in weeks) before fitting the BHM. Subjects' discount factors were highly correlated across the two tasks (Pearson $r = 0.92$; Spearman $r = 0.92$, all $p < 0.01$). Moreover, there is a high degree of overlap in the population estimates of $\log(k)$ for the two tasks (**Figure 6B**). If subjects had ignored units then we would expect that $\log(k_W) = \log(k_D) + \log(7) = \log(k_D) + 1.95$. Comparing the posteriors with that predicted shift, we can say that the shift is highly unlikely ($p < 0.0001$). Nonetheless, the discount factors in the two tasks were not equal. We observed a kind of amplification of preferences: the impulsive subjects were more impulsive in days than weeks and the patient subjects were more patient in days than weeks (**Figure 6—figure supplement 1**). We do not have an explanation for this effect, but overall this control experiment is consistent with and extends our main results: subjects' time-preferences are reliable but context-dependent and the context dependence cannot be explained by subjects ignoring the units of time.

Having ruled out the possibility that subjects ignore units of time, we test our second potential explanation: that subjects make decisions based on a subjective delay that is context dependent. We reasoned that if choices are context dependent then it may take some number of trials in each task before the context is set. Consistent with this reasoning, we found a small but significant adaptation effect in early trials in our main experiment: subjects are more likely to choose the later option

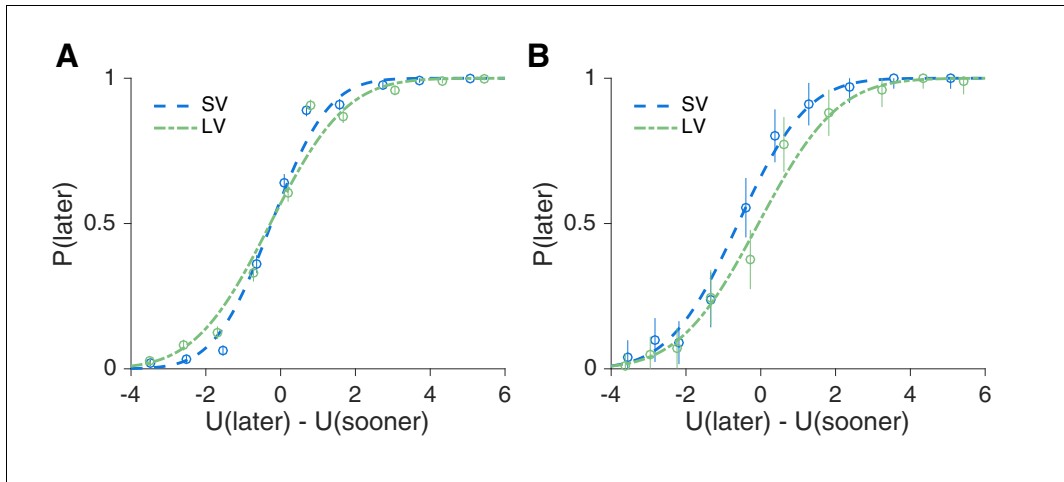

**Figure 7.** Evidence for context dependent temporal processing. (**A,B**) Main experiment early trials adaptation effect. The offers for each subject were converted into a subjective utility, U, based on the subjects' discount factors in each task. This allowed us to combine data across subjects to plot psychometric curves of the probability of choosing the later option, *P(later)*, for SV and LV averaged across all subjects comparing late trials (Trial in task > 5) (**A**) to the first four trials (**B**). Using a generalized linear mixed effects model, we found a significant interaction between *early/late* and *SV/LV* ($\beta_{SVLV:early} = 0.86 \pm 0.17, p < 10^{-6}, n_{subjects} = 63, n_{trials} = 20387$).
DOI: https://doi.org/10.7554/eLife.39656.023

in the first trials of SV task (*Figure 7A,B*). It seems that, at first, *seconds* in the current task are interpreted as being smaller than *days* in the preceding task, but within several trials *days* are forgotten and time preferences adapt to a new time-horizon of *seconds*.

## Discussion

We set out to test whether the same delay-discounting process is employed regardless of the verbal/non-verbal nature of the task and the time-horizon. We found significant correlations between subjects' discount factors across the three tasks, providing evidence that there are common cognitive (and presumably basal neural) mechanisms underlying the decisions made in the three tasks. In particular, the strong correlation between the short time-horizon non-verbal and verbal tasks ($r = 0.79$, *Figure 3A*) provides the first evidence for generalizability of the non-verbal task; suggesting that this task can be applied to both human and animal research for direct comparison of cognitive and neural mechanisms underlying delay-discounting. However, the correlation between the short-delay/non-verbal task and the long-delay/verbal task, while significant, is weaker ($r = 0.40$). Taken together, our results suggest animal models of delay-discounting may have more in common with short time-scale consumer behavior such as impulse purchases and 'paying-not-to-wait' in mobile gaming (*Evans, 2016*) and some caution is warranted when reaching conclusions from the broader applicability of these models to long-time horizon real-world decisions, such as buying insurance or saving for retirement.

### Reliability of preferences

The question of reliability is of central importance to applying in-lab studies to real-world behavior. There are several concepts of reliability that our study addresses. First, is test/re-test reliability; second, reliability across the verbal/non-verbal gap; third, reliability across the second/day gap. Consistent with previous studies (*Lane et al., 2003*; *Meier and Sprenger, 2015*; *Augenblick et al., 2015*), we found high test/re-test reliability. Choices in the same task did not differ when made at the beginning or the end of the session nor when they were made in sessions held on different days even 2 weeks apart.

We found a high degree of reliability in time-preferences across the verbal/non-verbal gap ($r = 0.79$, *Figure 3A*, *Table 1*, *Table 2*). This reliability has not been, to the best of our knowledge,

previously measured and is of similar strength to the reported test-retest reliability of personality traits (*Viswesvaran and Ones, 2000*; *Berns et al., 2007*). The closest literature that we are aware finds that value encoding (the convexity of the utility function) but not probability weighting is similar across the verbal/non-verbal gap in sessions that compare responses to a classic verbal risky economic choice task with an equivalent task in the motor domain (*Wu et al., 2009*). It may be that unlike time or value, probability is processed differently in verbal vs. non-verbal settings (*Hertwig and Erev, 2009*). The main difference between choices in the NV and SV tasks was the increase in noise in NV. A worthwhile future direction is to disentangle the neural substrates of perceptual noise vs. decision noise in a non-verbal task of economic preferences (*Hanks et al., 2015*; *Constantinople et al., 2018*).

We found a moderate degree of reliability across the second/day gap ($r = 0.61$, *Figure 3B*, *Table 1*, *Table 2*). There are several aspects to the time-horizon gap that may contribute independently to the lower correlations observed between our short and long tasks (compared to the two short tasks). First, there is the difference in order of magnitudes of the delays. Second, there is a difference in the experience of the delayed rewards, in that subjects must wait, staring at the clock, through all delays in the short tasks, but in the long task, subject wait for a single reward, but can go about their lives while waiting. *Paglieri (2013)* described these as 'waiting' in seconds compared to 'postponing' in days. Third, our short tasks had a 'coin drop' sound at the time of the reward, which may have acted as a secondary reinforcer and contributed to the discounting of delayed rewards. The absence of this from the long task may have contributed to the decreased reliability between short and long tasks.

Our control study using delays of days vs. weeks compared tasks with different scales but did not differ in the experience of the delayed rewards, as in LV, only (at most) one delayed reward was experienced for both days and weeks tasks. In that experiment, we found extremely high reliability between time-preferences across tasks (*Figure 6a*). That is, *Figure 6* shows that on average, subjects discounted 7 days as frequently as 1 week was discounted in the other task. While, days and weeks are only scaled by seven times and may be easily approximated via preexisting rules of thumb, seconds vs. days are scaled by 86400. Moreover, people have more practice at converting days and weeks than seconds and days. So while the days/weeks experiment provides some evidence that a difference in the magnitude of the delays does not, on its own, affect reliability, it may be that larger or unfamiliar differences (e.g. an experiment comparing hours vs. weeks) may do so. Still, we find the second hypothesis for the lower reliability across time-horizons more compelling: that individual differences in subjective costs of waiting are distinct from (but correlated with) individual differences in costs of postponing (discussed in more detail below).

The evidence from the literature on the issue of reliability across time-horizons is mixed. On the one hand, some have found that measures of discount factors on month-long delays are not predictive of discount factors for year-long horizons (a difference of one order of magnitude) (*Thaler, 1981*; *Loewenstein and Thaler, 1989*) but others have found consistent discounting for the same ranges (*Johnson and Bickel, 2002*). Other studies that compared the population distributions of discount factors for short (up to 28 days) to long (years) delays (2 orders of magnitude) found no differences in subjects' discount factors (*Eckel et al., 2005*; *Andersen et al., 2014*). Some of these discrepancies can be attributed to the framing of choice options: standard larger later vs. smaller sooner compared to negative framework (*Loewenstein and Thaler, 1989*), where subjects want to be paid more if they have to worry longer about some negative events in the future.

Several previous studies have compared discounting in experienced delay tasks (as in our short tasks) with tasks where delays were hypothetical or just one was experienced (*Johnson and Bickel, 2002*; *Lane et al., 2003*; *Reynolds and Schiffbauer, 2004*; *Navarick, 2004*; *Horan et al., 2017*). For example, *Lane et al. (2003)*, also used a within-subject design to examine short vs. long delays (e.g. similar to our short-verbal and long-verbal tasks) and found similar correlations ($r \sim 0.5 \pm 0.1$) with a smaller sample size (n = 16). (Interestingly, they also found, but did not discuss, a 5 order of magnitude scaling factor between subjects' discounting of seconds and days suggesting that this is a general phenomenon.)

## Subjective scaling of time

It may seem surprising that human subjects would discount later rewards, that is choosing small immediate rewards, in a task where delays are in seconds. After all, subjects cannot consume

earnings immediately. Yet, this result is consistent with earlier work that suggests individuals derive utility from receiving money irrespective of when it is consumed (*Reuben et al., 2010*; *McClure et al., 2004*; *McClure et al., 2007*). In our design, a pleasing (as reported by subjects) 'slot machine' sound accompanied the presentation of the coins in the short-delay tasks. This sound may be experienced as an instantaneous secondary reinforcer (*Kelleher and Gollub, 1962*). Whether or not the secondary reinforcer used in our task is experienced in an analogous way to primary rein-forcers used in animal studies may limit the degree of overlap in underlying neural mechanisms. On the other hand, our subjects' behavior would not be surprising for those who develop (or study) 'pay-not-to-wait' video games (*Evans, 2016*), which exploit player's impulsivity to acquire virtual goods with no actual economic value.

The range of rates of discounting we observed in the long-verbal task was consistent with that observed in other studies. For example, in a population of more than 23,000 subjects the log of the discount factors ranged from −8.75 to 1.4 (*Sanchez-Roige et al., 2018*), which is similar to the ranges presented in *Figure 3B*. This implies that, in our short tasks, subjects are discounting extremely steeply, that is they are discounting the rewards *per second* at approximately the same amount that they discounted the reward *per day*. This discrepancy has been previously found (*Lane et al., 2003*; *Navarick, 2004*). We consider three (non-mutually exclusive) explanations for this scaling. First, subjects may ignore units. However, by testing overlapping time-horizons of days and weeks we confirmed that subjects can pay attention to units.

Second, it may be that with short delay tasks we are capturing cost of waiting while long delay tasks measure delay-discounting. The costs of waiting could take several forms (*Paglieri, 2013*). One form is the cost of boredom (*Mills and Christoff, 2018*); a feeling which animals may also experi-ence (*Wemelsfelder, 1984*). Subjects could find it painful to sit and wait, staring at the clock on the computer screen, during the delay. Additionally, there could be opportunity costs related to how much subjects value their own time. We found that in the short tasks, subjects with large discount factors also performed the task faster (*Figure 3—figure supplement 1*). If these subjects value their time more and thus have higher costs of waiting, then given our results *Figure 3B* there is a surpris-ingly large correlation between how much subjects value their time (in the short tasks) and how much they discount postponed rewards (in the long task). Regardless of the precise form of the costs of waiting (*Chapman, 2001*; *Paglieri, 2013*; *Navarick, 2004*) in order for these costs to explain the temporal scaling we observed between short and long tasks, relative to the costs of postponing, they would have to be, coincidentally, close in value to the number of seconds in a day.

We feel this coincidence is unlikely, and thus favor the third explanation for the scaling: temporal context. When making decisions about seconds, subjects 'wait' for seconds and when making deci-sions about days subjects 'postpone reward' for days (*Paglieri, 2013*). Although our experiments were not designed to test whether the strong effect of temporal context was due to normalizing, existence of extra costs for waiting in real time, or both, we did find some evidence for the former (*Figure 6C*). Consistent with this idea, several studies have found that there are both systematic and individual level biases that influence how objective time is mapped to subjective time for both short and long delays (*Wittmann and Paulus, 2009*; *Zauberman et al., 2009*). Thus, subjects may both normalize delays to a reference point and introduce a waiting cost at the individual level that will lead short delays to seem as costly as the long ones.

## Conclusion

We have shown for the first time that there is a high degree of reliability across verbal and non-ver-bal delay-discounting tasks. In the analysis of experimental data, we found several interesting phe-nomena which warrant further examination at both the behavioral and the neural level: the extreme scaling effects from seconds to days; the compression toward the mean of discount factors in weeks vs. days; and the adaptation observed at the beginning of tasks. Nonetheless, these effects were consistent across the subject population: affecting the quantitative estimate of discount factor but not the subjects' impulsivity relative to the group. Overall, this work provides support for connecting non-verbal animal studies with verbal human studies of delay-discounting.

## Materials and methods

### Participants

For the main experiment, participants were recruited from the NYU Shanghai undergraduate student population on two occasions leading to a total sample of 67 (45 female, 22 male) NYU Shanghai students. Using posted flyers, we initially recruited 35 students but added 32 more to increase statistical power (the power analysis indicates that for expected correlation $r = 0.5$ and 80% power (the ability of a test to detect an effect, if the effect actually exists; *Cohen, 1988*; *Bonett and Wright, 2000*) the required sample size is N = 29, for a medium size correlation of $r = 0.3$ the required sample size is N = 84).

The study was approved by the IRB of NYU Shanghai. The subjects were between 18–23 years old, 34 subjects were Chinese Nationals (out of 67). They received a 30 CNY (~$5 USD) per hour participation fee as well as up to an additional 50 CNY (~$8 USD) per session based on their individual performance in the task (either in NV task, or total in SV and LV tasks, considering the delay of payment in the LV task). The experiment involved five sessions per subject (three non-verbal sessions followed by two verbal sessions), permitting us to perform within-subject analyses. The sessions were scheduled bi-weekly and took place in the NYU Shanghai Behavioral and Experimental Economics Laboratory. In each session, all decisions involved a choice between a later (delay in seconds and days) option and an immediate (now) option. Three subjects did not pass the learning stages of the NV task. One subject did not participate in all of the sessions. These four subjects were excluded from all analyses.

### Experimental design

The experiments were constructed to match the design of tasks used for rodent behavior in Prof. Erlich's lab. We provided relatively minimal instructions for the subjects other than explaining that coins were worth real money (See subject instructions in *Supplementary file 4* and *Supplementary file 5*). For the temporal discounting task, the value of the later option is mapped to the frequency of pure tone (frequency $\propto$ reward magnitude) and the delay is mapped to the amplitude modulation (modulation period $\propto$ delay). The immediate option was the same on all trials for a session and was unrelated to the sound. There were 25 different 'later' options presented in each task: all possible combinations of 5 delays (3, 6.5, 14, 30, 64) and 5 reward magnitudes (1, 2, 5, 8, 10). The immediate option was fixed at 4 coins, so later offers of 1 or 2 were considered 'smaller-later' offers that were created to encourage subjects to pay attention to the sound in the non-verbal task, and to make sure subjects were paying attention in the verbal tasks. In the non-verbal task the two 'smaller-later' options made up 25% of later options, whereas in the verbal experiment they made up 10% of later options. All 'larger-later' offers were equally likely to be presented. Given that the smaller later option is always strictly worse than larger immediate option, if in such a trial 'smaller-later' is chosen, economic theory would classify this choice as reflecting a first-order violation. The offers in each task were structured in short blocks. Each block used the same reward magnitude for a 'later' option offered at different randomly ordered delays. For each subject the order of reward blocks was chosen randomly. Jittering the number of trials in each reward resulted in 160 trials on average for each subject in each of the two verbal tasks and up to 200 trials in the non-verbal task.

Through experiential learning, subjects learned the map from visual and sound attributes to values and delays. This was accomplished via six learning stages (0, 1, 2, 3, 4, 5; *Video 1*) that build up to the final non-verbal task (NV) that was used to estimate subjects' discount-factors. Briefly, the first four stages were designed

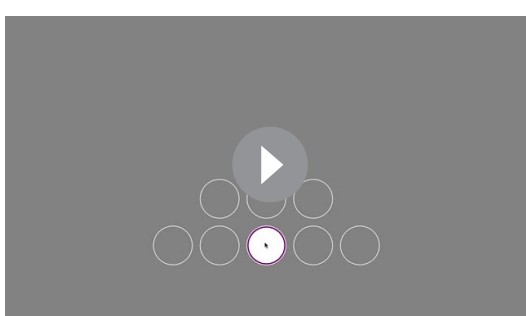

**Video 1.** Learning. A video of the learning stages, showing the examples of violations that can be made. The video starts with stage 0 and continues with stage 1 at 00:14, stage 2 at 00:31, stage 3 at 00:55, stage 4 (trimmed) at 01:18 and stage 5 (trimmed) at 01:41.
DOI: https://doi.org/10.7554/eLife.39656.024

to (0) learn that a mouse-click in the middle bottom 'reward-port' produced coins (that subjects knew would be exchanged for money), (1) learn to initiate a trial by a mouse-click in a highlighted port, (2) learn 'fixation': to keep the mouse-cursor in the highlighted port, (3) associate a mouse-click in the blue port with the sooner option (a reward of a fixed 4 coin magnitude that is received instantly) (4) associate varying tone frequencies with varying reward at the yellow port (5) associate varying amplitude modulation frequencies with varying delays at the yellow port. In stage 4, subjects are primed to the sound frequency to learn the variability of reward magnitudes: first, the lower and upper bounds, then, in ascending and descending order and, finally, in random order. In the final stage 5, subjects heard the AM of a sound during fixation that is now mapped to the delay of the later option. The order of the stimuli presented was the same as in the previous stage. On each trial of the stage 3, 4 and 5 there was either a blue port or a yellow port (but not both). The exact values for reward and delay parameters experienced in the learning stages correspond to values that are used throughout the experiment. After selecting the yellow-port (i.e. the delayed option), a count-down clock appeared on the screen and the subject had to wait for the delay which had been indicated by the amplitude modulation of the sound for that trial. Any violation (i.e. a mouse-click in an incorrect port or moving the mouse-cursor during fixation) was indicated by flashing black circles over the entire 'poke' wall accompanied by an unpleasant sound (for further demonstration of the experimental time flow, please see the *Video 1*).

When a subject passed the learning stages (i.e. four successive trials without a violation in each stage, *Figure 1—figure supplement 1*), they progressed to the decision stages of the non-verbal task (NV). Progressing from the learning stages, a two-choice decision is present where the subject can choose between an amount now (blue choice) versus a different amount in some number of seconds (yellow choice). During the decision stages the position of blue and yellow circles on the poke wall was randomized between left and right and was always symmetrical (*Figure 1*, *Video 2*). Each of the three non-verbal sessions began with learning stages and continued to the decision stages. In the 2nd and the 3rd non-verbal sessions, the learning stages were shorter in duration.

The final two sessions involved verbal stimuli (*Video 3*, *Video 4*). During each session, subjects experience an alternating set of tasks: short delay (SV)–long delay (LV)–SV–LV (or LV-SV-LV-SV, counter-balanced per subject). An example of a trial from the short time-horizon task (SV) is shown in the sequence of screens presented in *Figure 1*. The verbal task in the long time-horizon (LV) includes Initiation, Decision (as in *Figure 1*) and the screen that confirms the choice. There are two differences in the implementation of these sessions relative to the non-verbal sessions. First, the actual reward magnitude and delay are written within the yellow and blue circles presented on the screen, in place of using sounds. Second, in the non-verbal and verbal short delay sessions, subjects continued to accumulate coins (following experiential learning stages) and the total earned was paid via electronic payment at the end of each experimental session. In the long-verbal sessions, a single trial was selected at random at the conclusion of the session for payment (method of payment commonly used in human studies with long delays, (*Cox and Kable, 2014*)). The associated payment is made now or later depending on the subject's choice in the selected trial.

## Control experiment 1: No Circles (NC)

In total, 25 (29 started, 4 withdrew) undergraduate students from NYU Shanghai participated in five experimental sessions (three non-verbal and two verbal sessions, in this sequence, that were scheduled bi-weekly). The study requirements in order to meet the IRB protocol conditions remained the same as in the main experiment. In each session, subjects completed a series of intertemporal choices. Across sessions, at least 160 trials were conducted in each of the following tasks mimicking the main experiment, (i) non-verbal (NV), (ii) verbal short delay (SV, 3–64 s), and (iii) verbal long delay (LV, 3–64 days). In each trial, irrespective of the task, subjects made

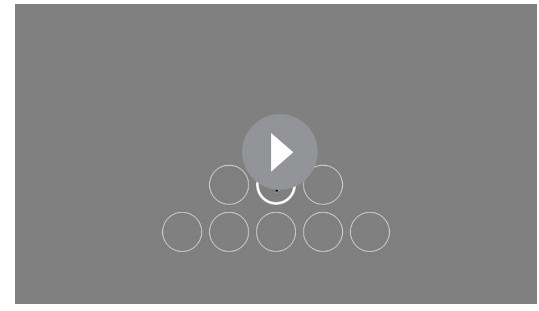

**Video 2.** NV. A video of the several consecutive trials of the non-verbal task.
DOI: https://doi.org/10.7554/eLife.39656.025

a decision between the sooner and the later options. The NV task was exactly the same as in the main experiment. All subjects passed learning stages. The SV and LV tasks differed from the main experiment in exactly two ways: First, the stimuli presentation didn't include a display of circles of different colors. Instead, two choices were presented on the left or on the right side (counterbalanced) of the screen (*Figure 4—figure supplement 1*); Second, the subjects did not have to click on the circles using mouse, instead they used a keyboard to indicate 'L' or 'R' choice. Everything else stayed the same as in the main experiment, that is the last two sessions included an alternating set of verbal tasks: SV-LV-SV-LV (or LV-SV-LV-SV, for a random half of subjects), the payment was done differently for SV and LV (randomly picked trial for payment in LV), etc. The purpose of this control experiment is to confirm that significant correlation between non-verbal tasks and verbal tasks we report in Results is not an artifact of our main experimental design: subjects experience the same visual display and motor responses in the non-verbal and verbal tasks and this design similarity might drive the correlation between time-preferences in these tasks. Instead, in this control experiment the verbal tasks are made as similar as possible (keeping our experiment structure) to typical intertemporal choice tasks used in human subjects.

## Control experiment 2: Days and weeks (DW)

In total, 16 subjects took part in this experiment (2 of 16 were excluded from analyses because their choices were insensitive to delay). Subjects were undergraduate students from NYU Shanghai. This experiment was approved under the same IRB protocol as the control experiment 1 and the main experiment. This experiment included two following experimental tasks: (i) verbal days delay (DV, 1–64 days) and (ii) verbal weeks delay (WV, 1–35 weeks). Subjects underwent only one session where the verbal tasks were alternated: DV-WV-DV-WV (or WV-DV-WV-DV, for roughly half of subjects; 200 trials per task). For each of the tasks in this control experiment the stimuli and procedures were exactly the same as for LV task in the control experiment 1. The purpose of this control task is to check whether subjects pay attention to units.

## Significance tests of demographic and psychological categories

We did not find any significant differences between any of the categorical subjects' groups, including gender and nationality in learning stages (*Figure 1—figure supplement 1*), intertemporal decisions and first-order violations. For the proportion of 'yellow' choice (mean $\pm$ std. dev.) there is no significant difference between females and males (females: $0.56 \pm 0.24$ males: $0.53 \pm 0.29$ Wilcoxon rank sum test, $p = 0.22$) and between Chinese and Non-Chinese (Chinese: $0.57 \pm 0.23$ Non-Chinese: $0.54 \pm 0.29$ Wilcoxon rank sum test, $p = 0.33$) subjects. Similarly, for the first-order violations there is no significant difference between females and males (violations per session, females: $1.14 \pm 1.97$ males: $1.07 \pm 2.20$ Wilcoxon rank sum test, $p = 0.2607$) and a slight difference between Chinese and Non-Chinese (Chinese: $1.21 \pm 2.07$ Non-Chinese: $1.00 \pm 2.00$ Wilcoxon rank sum test, $p < 0.1$) subjects.

We used the Barratt Impulsiveness Scale (BIS-11; (*Patton et al., 1995*)) as a standard measure of impulsivity. This test is reported to often correlate with biological, psychological, and behavioral characteristics. The mean total score for our students sample was 61.79 (std = 9.53), which is consistent with other reports in the literature (e.g., (*Stanford et al., 2009*)). The BIS-11 did not correlate significantly with the estimated discount factors (BIS vs. $\log(k_{NV})$: Pearson $r = 0.2$, $p = 0.1180$; BIS vs. $\log(k_{SV})$: Pearson $r = 0.19$, $p = 0.1384$; BIS vs. $\log(k_{LV})$: Pearson $r = 0.15$, $p = 0.2521$). Prior research finds mixed evidence of the association between the BIS-11 and delay: some report significant positive correlations (*Mobini et al., 2007*; *Beck and Triplett, 2009*; *Cosenza and Nigro, 2015*), others do not find significant correlations and suggest that delay-discounting tasks might measure a different aspect of impulsivity (*Mitchell, 1999*; *Fellows and Farah, 2005*; *Reynolds et al., 2006*; *Saville et al., 2010*). Following earlier research that reports that components of the BIS score might drive the correlation with discounting (*Fellows and Farah, 2005*; *Mobini et al., 2007*; *Beck and Triplett, 2009*; *Ahn et al., 2016*), we next decomposed the score. Similar to others we found that correlation between BIS nonplanning component and $\log(k_{NV})$ is positive and significant (Pearson $r = 0.3$, $p < 0.05$).

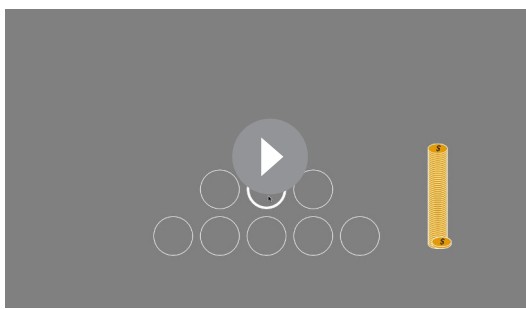

**Video 3.** SV. A video of the several consecutive trials of the short delay task.
DOI: https://doi.org/10.7554/eLife.39656.026

## Time and reward re-scaling

In our main experimental tasks we used two units for delays: seconds and days, where 1 day = 86400 s. We also used three exchange rates: for non-verbal task 1 coin = 0.1 CNY; for verbal short delay 1 coin = 0.05 CNY; for long delay 1 coin = 4 CNY. Humans tend to discount large rewards less steeply than small rewards, that is discounting rates tend to increase as amounts decrease (*Green et al., 1999*; *Green et al., 2004*). We re-calculated the model-based (softmax-hyperbolic model) median BHM model fits: 1) we convert them to the same units (1/days): $k_{NV} = 4173.1$ (by multiplying $k \sim 1/day$ by the day to seconds conversion rate), $k_{SV} = 2548.8$, $k_{LV} = 0.0356$, 2) we consider reward re-scaling: "going from \$10 to \$.20, a factor of 50, k values would increase by a factor of 2" (*Navarick, 2004*) $k_{NV} = 4173.1$, $k_{SV} = 2548.8$, $k_{LV} = 0.0712$ and 3) conclude that discrepancy of discount rates between time-horizons cannot be accounted by magnitude effects. Thus, the discount rate revealed in the verbal short delay task is more than $10^4$ times larger than the rate describing the choices made by the same participants in the verbal long delay task.

## Analysis

In order to be sure that our results and main conclusions did not depend on the method (e.g. Bayesian hierarchical vs. maximum likelihood estimation of individual subject parameters) or functional form (e.g. exponential vs. hyperbolic), we validated our results with several methods. We estimated subjects' time-preferences individually (since discounting factors differ among people) for each experimental task with maximum likelihood estimation (MLE) and used leave-one-trial-out cross-validation for model comparison. In the delay-discounting literature, there is no consensus which functional form of discounting best describes human behavior: the exponential model (*Samuelson, 1937*) of time discounting has a straightforward economic meaning - a constant probability of loss of reward per waiting time, whereas the hyperbolic model (*Mazur, 1987*) seems to more accurately describe how individuals discount future rewards, in particular preference reversals (*Berns et al., 2007*). We considered both a shift-invariant softmax rule and a scale-invariant matching rule to transform the subjective utilities of the sooner and later offers into a probability of choosing the later offer. Thus, we considered four model classes: (1) hyperbolic utility with softmax, (2) exponential utility with softmax, (3) hyperbolic utility with matching rule and (4) exponential utility with matching rule. We also considered models that account for utility curvature, that is $V$ is replaced by $V^{\alpha_i}$ and models that account for trial number and cumulative waiting time. Based on the Bayesian information criterion criterion (BIC; top three models by BIC: (2) $-179.47$ (SE = 4.99), (1) $-191.16$ (SE = 4.96), and (4) $-192.03$ (SE = 4.82)) and number of subjects that were well described by the models, the softmax-hyperbolic model (1) was selected.

Following modern statistical convention, we used a Bayesian hierarchical model (BHM) brms, 2.0.1 (*Carpenter et al., 2016*; *Bürkner, 2017*) that allows for pooling data across subjects, recognizing individual differences and estimating posterior distributions, rather than point estimates of the parameters. We validated that our results were not sensitive to the model fitting methods used; the means of BHM posteriors of the individual discount-factors for each task are almost identical to the individual fits done for

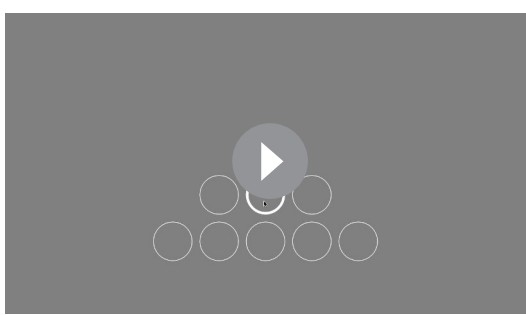

**Video 4.** LV. A video of the several consecutive trials of the long delay task.
DOI: https://doi.org/10.7554/eLife.39656.027

each experimental task separately using maximum likelihood estimation through fmincon in Matlab (*Figure 2—figure supplement 1* , *Figure 2—figure supplement 2*). We further validated the BHM method by simulating choices from a population of 'agents' with known parameters and demonstrating that we could recover those parameters given the same number of choices per agent as in our actual dataset (not shown). In order to assess the goodness of fit for individual subjects in each task, we computed the Bayesian $r^2$ using the 'bayes_R2' function in the 'brms' package in R.

The first non-verbal session data was excluded from model-fitting due to a comparatively high proportion of first-order violations relative to the following two non-verbal sessions (from 26% of trials in the first non-verbal session ($NV_1$) to 19% and 13% for the next two non-verbal sessions, $NV_2$ and $NV_3$, respectively, Wilcoxon signed-rank test, $NV_1$ vs. $NV_2$ & $NV_1$ vs. $NV_3$, $p < 0.01$). Compliance with first-order stochastic dominance means that, in principle, this observed behavior can be adequately modeled with a utility-function style analysis (*Tymula et al., 2013*; *Yamada et al., 2013*). In the non-verbal task, violations could result from lapses in attention, motor errors or difficulty in transforming the perceptual stimuli into offers (in particular, early on in the first session while learning has not completed). In the verbal tasks, inattention and/or misunderstanding are likely explanations of violations. It is important that $NV_1$ did not differ significantly in choice consistency (the number of preference reversals was not significantly different between $NV_1$ and later non-verbal sessions, Wilcoxon signed-rank test, all $p > 0.2$).

A six population level and four subject level parameters model (mixed-effects model) is used to estimate discount-factors and decision-noise from choices. Using the 'brms' (*Bürkner, 2017*) package in R allows to do BHM of nonlinear multilevel models in Stan (*Carpenter et al., 2016*) with the standard R formula syntax:

$$\text{choice} \sim$$

$$\text{inv\_logit}((\text{later\_reward}/(1+\textbf{exp}(\text{logk})*\textbf{delay}) - \text{sooner\_reward})/\text{noise}),$$

$$\text{noise} \sim \text{task} + (1\,|\,\text{subjid}),$$

$$\text{logk} \sim \text{task} + (\text{task}\,|\,\text{subjid})$$

where later_reward is the later reward, sooner_reward is the sooner reward; logk is the natural logarithm of the discounting parameter $k$ and noise ($\tau$) is the decision noise (as in *Equation 1* and *Equation 2*, respectively). The population level effects estimate shared shifts in delay discounting $\log(k)$ and decision noise $\tau$ (e.g. if all subjects are more impulsive in one task).

At the subject level, this model transforms the reward and delays on each trial and individual preferences into a probability distribution about the subject's choice. For the non-verbal task, we assumed that the subjects had an unbiased estimate of the meaning of the frequency and AM modulation of the sound. Rewards and delays are converted in the subjective value of each choice option using hyperbolic utility model (*Equation 1*). Then, *Equation 2* (a logit, or softmax function) translates the difference between the subjective value of the later and the subjective value of the sooner (estimated using *Equation 1*) into a probability of later choice for each subject. Two functions below rely on the four parameters ($k_{i,t}$: ($k_{i,NV}, k_{i,SV}, k_{i,LV}$), the discounting factor per subject, $i$, in each task, $t$, and $\tau_{i,t}$ individual decision noise). For example, for subject 12 in task $NV$ the effective discount factor is the product of the population level discount factor in $NV$ and subject 12 effect in $NV$, $k_{12,NV} = \hat{k}_{NV} \times \dot{k}_{12,NV}$.

Hyperbolic utility model:

$$U_i = \frac{V}{1 + k_{i,t}T} \tag{1}$$

where $V$ is the current value of delayed asset and $T$ is the delay time.

Softmax rule:

$$P(Li) = \frac{e^{U_{Li}/\tau_i}}{e^{U_{Li}/\tau_i} + e^{U_{Si}/\tau_i}} \qquad (2)$$

where $L$ is the later, $S$ is the sooner offer and $\tau_i$ is the individual decision noise.

To test for differences across tasks we examined the BHM fits using the 'hypothesis' function in the 'brms' R package. This function allows us to directly test the posterior probability that the $\log(k)$ is shifted and/or scaled between treatments. This function returns an 'evidence ratio' which tells us how much we should favor the hypothesis over the inverse (e.g. $\frac{P(a>b)}{P(a<b)}$) and we used Bayesian confidence intervals to set a threshold ($p < 0.05$) to assist frequentists in assessing statistical significance.

The bootstrapped (mean, median and variance) tests are done by sampling with replacement and calculating the sample statistic for each of the 10000 boots, therefore creating a distribution of bootstrap statistics and (i) testing where 0 falls in this distribution for unpaired tests or (ii) doing a permutation test to see whether the means are significantly different for paired tests.

Simulations done for both model-based and model-free analyses are described in detail in (*Figure 2—figure supplement 3* , *Figure 3—figure supplement 4*).

To estimate the effect of adaptation (*Figure 6B,C*), we first used the fitted parameters from the hierarchical model to transform each offer to each subject into a difference in utility, $\Delta U$. We classified the first four trials in a long or short task as *early* trials. Then, we fit a generalized linear mixed model (using the function 'glmer' from the 'lme4' R package) where we fit the choice of the subjects with fixed-effects $\Delta U$, *early/late*, *LV/SV*, and interactions between *LV/SV:$\Delta U$* and *early/late:LV/SV*. We also included a slope and intercept for each subject as random effects. To test for the significance of this adaptation effect, we compared this model to a reduced nested model where we removed the *early/late* term and interaction *early/late:LV/SV*.

Full model with adaptation:

$$choice \sim \Delta U + LV\,SV : \Delta U + LV\,SV * early + (1 + \Delta U | subjid)$$

Reduced model without adaptation:

$$choice \sim \Delta U + LV\,SV : \Delta U + LV\,SV + (1 + \Delta U | subjid)$$

|  | Df | AIC | BIC | logLik | Deviance | Chisq | Chi df | Pr(>chisq) |
|---|---|---|---|---|---|---|---|---|
| **Reduced Model** | 7 | 10955.51 | 11010.97 | −5470.76 | 10941.51 | | | |
| **Full Model** | 9 | 10932.40 | 11003.70 | −5457.20 | 10914.40 | 27.11 | 2 | $<10^{-4}$ |

## Software

Tasks were written in Python using the PsychoPy toolbox (1.83.04, (*Peirce, 2007*)). All analysis and statistics was performed either in Matlab (version 8.6, or higher, The Mathworks, MA), or in R (3.3.1 or higher, R Foundation for Statistical Computing, Vienna, Austria). R package 'brms'(2.0.1) was used as a wrapper for Rstan (*Guo et al., 2016*) for Bayesian nonlinear multilevel modeling (*Bürkner, 2017*), shinystan (*Gabry, 2015*) was used to diagnose and develop the brms models. R package 'lme4' was used for linear mixed-effects modeling (*Bates et al., 2014*).

## Data availability

Software for running the task, as well as the data and analysis code for regenerating our results are available at https://github.com/erlichlab/delay3ways/tree/v1.0 (*Lukinova and Erlich, 2018*; copy archived at https://github.com/elifesciences-publications/delay3ways).

## Acknowledgements

We thank NYU Shanghai undergraduate students Stephen Mathew, Xirui Zhao, Wanning Fu and Jonathan Lin and Research Specialist at Behavior and Experimental Economics Lab Siyan Yao who helped with data collection. Members of the Erlich lab provided thoughtful feedback throughout the project. Paul Glimcher, Ming Hsu and Joseph Kable contributed helpful advice about this

project. We also thank Mehrdad Jazayeri and the other reviewers for their suggestions which substantially improved the paper through the review process.

## Additional information

### Funding

| Funder | Grant reference number | Author |
| --- | --- | --- |
| National Science Foundation of China | NSFC-31750110461 | Evgeniya Lukinova |
| Shanghai Eastern Scholar Program | | Evgeniya Lukinova |
| NYU Shanghai | Research Challenge Grant | Steven F Lehrer Jeffrey C Erlich |
| Science and Technology Commission of Shanghai Municipality | 15JC1400104 | Jeffrey C Erlich |
| NYU-ECNU Joint Institute for Brain and Cognitive Science at NYU Shanghai | | Jeffrey C Erlich |

The funders had no role in study design, data collection and interpretation, or the decision to submit the work for publication.

### Author contributions

Evgeniya Lukinova, Jeffrey C Erlich, Conceptualization, Resources, Data curation, Software, Formal analysis, Supervision, Funding acquisition, Validation, Investigation, Visualization, Methodology, Writing—original draft, Project administration, Writing—review and editing; Yuyue Wang, Software, Investigation; Steven F Lehrer, Conceptualization, Resources, Supervision, Funding acquisition, Writing—review and editing

### Author ORCIDs

Evgeniya Lukinova http://orcid.org/0000-0002-8357-9307
Jeffrey C Erlich http://orcid.org/0000-0001-9073-7986

### Ethics

Human subjects: The study was approved by the institutional review board of NYU Shanghai following all Chinese and USA regulations regarding human subjects research (IRB Protocol #003-2015). All subjects were NYU Shanghai students recruited on campus and gave informed consent and consent to publish the results (with anonymized data) before participation in the study.

### Decision letter and Author response

Decision letter https://doi.org/10.7554/eLife.39656.036
Author response https://doi.org/10.7554/eLife.39656.037

## Additional files

### Supplementary files

• Supplementary file 1. Individual subjects fits for main experiment. Each plot is the softmax-hyperbolic fit for each subject in the main experiment. In each panel, the marker and error bar indicate the mean and binomial confidence intervals of the subjects choices for that offer. The smooth ribbon indicated the BHM model fits (at 50, 80, 99% credible intervals). At the top of each subject plot we indicate the mean estimates of $\log(k)$ and $\tau$ for each task for that subject. We also indicate the Bayesian $r^2$ for each task. Plots from Left to right, row-by-row are ordered by discount factor (as estimated using BHM) for SV.

DOI: https://doi.org/10.7554/eLife.39656.029

• Supplementary file 2. Individual subjects fits for control experiment 1. Each plot is the softmax-hyperbolic fit for each subject in the control experiment 1. In each panel, the marker and error bar indicate the mean and binomial confidence intervals of the subjects choices for that offer. The smooth ribbon indicated the BHM model fits (at 50, 80, 99% credible intervals). At the top of each subject plot we indicate the mean estimates of $\log(k)$ and $\tau$ for each task for that subject. We also indicate the Bayesian $r^2$ for each task. Plots from Left to right, row-by-row are ordered by discount factor for SV.

DOI: https://doi.org/10.7554/eLife.39656.030

• Supplementary file 3. Individual subjects fits for control experiment 2. Each plot is the softmax-hyperbolic fit and data for each subject in control experiment 2. In each panel, the marker and error bar indicate the mean and binomial confidence intervals of the subjects choices for that offer. The smooth ribbon indicated the BHM model fits (at 50, 80, 99% credible intervals). At the top of each subject plot we indicate the mean estimates of $\log(k)$ and $\tau$ for each task for that subject. We also indicate the Bayesian $r^2$ for each task. Plots from Left to right, row-by-row are ordered by discount factor for SV.

DOI: https://doi.org/10.7554/eLife.39656.031

• Supplementary file 4. Subject Instructions for non-verbal task.
DOI: https://doi.org/10.7554/eLife.39656.032

• Supplementary file 5. Subject Instructions for verbal tasks.
DOI: https://doi.org/10.7554/eLife.39656.033

• Transparent reporting form
DOI: https://doi.org/10.7554/eLife.39656.034

## Data availability

Data and code is available on GitHub (https://github.com/erlichlab/delay3ways/tree/v1.0; copy archived at https://github.com/elifesciences-publications/delay3ways).

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
