## [Decision Letter]

[**Editorial note:** This article has been through an editorial process in which the authors decide how to respond to the issues raised during peer review. The Reviewing Editor's assessment is that all the issues have been addressed.]

Thank you for submitting your article "Time preferences are reliable across time-horizons and verbal vs. experiential tasks" for consideration by *eLife*. Your article has been reviewed by three peer reviewers, and the evaluation has been overseen by a Reviewing Editor and Timothy Behrens as the Senior Editor. The following individual involved in review of your submission has agreed to reveal his identity: Mehrdad Jazayeri (Reviewer #2). The other reviewers remain anonymous.

The Reviewing Editor has highlighted the concerns that require revision and/or responses, and we have included the separate reviews below for your consideration. If you have any questions, please do not hesitate to contact us.

Summary:

Presently, there is no universal consensus as to whether the discount rates estimated in humans and other animals using different behavioral paradigms of inter-temporal choice all concern the same underlying process or not. This study manipulated two important factors related to this important question, namely, how the delay is cued or indicated, and the range of temporal intervals between choice and reward delivery. The main results support that there is a common core mechanism.

Major concerns:

Reviewers raised several major concerns regarding the description of the methods and experimental design, and also proposed some additional control analyses.

Separate reviews (please respond to each point):

*Reviewer #1:*

Time preferences are investigated in both humans and animals, but temporal discounting tasks used in humans and animals differ with regard to duration of temporal delays and stimulus modalities. To empirically test the impact of these factors, the authors compared discount rates in humans between a novel non-verbal temporal discounting task with verbal tasks including short and long delays. The results reveal moderate to strong correlations between discount factors across tasks.

There is much to like about the well-written manuscript: it addresses the important topic of the relevance of animal studies for understanding human behavior, the three temporal discounting tasks were rigorously matched with regard to choice structure, and several control analyses and control experiments were conducted to test alternative hypotheses. My main concern is that the current results allow no clear conclusions with regard to the theoretical question which mechanisms are shared (and which not) by the different temporal discounting tasks and that the parallels may be more imposed than it currently seems.

Theoretical impact:

The goal of the current experiments was to assess the impact of two methodological differences between human and animal studies, time horizon and stimulus modality (language-based vs. language-free). However, as the authors themselves write in the Discussion section, comparing discount rates across short and long time delays is not really novel, as this issue has already between addressed by previous studies. This leaves the comparison of different stimulus modalities as the main novel aspect of the current study, and unfortunately the results are not very conclusive here. This is because the correlations of the SN with the SV (0.54) and LV (0.36) are moderate in effect size, which suggests that the tasks measure both partially overlapping and at the same time also clearly dissociable constructs. While this is an interesting empirical finding in its own right, the implications of such moderate correlations for comparisons between human and animal studies are less clear.

The authors themselves appear to oscillate between different interpretations: for example, according to the manuscript title time preferences are reliable across verbal and experiential tasks, and also the Discussion section acknowledges that the three tasks share common cognitive and neural mechanisms, whereas at the same time "caution is warranted" when reaching conclusions from experiential short-delay paradigms in animals to verbal long-delay intertemporal decisions is humans. All this is true and at the same time somewhat trivial. From a theoretical perspective it would have been informative to get some idea about what these common and dissociable mechanisms might be. However the current study provides no answer to this question and further experiments would be needed.

Methodological concerns:

The authors used the monetary rewards in the different temporal discounting tasks, but it seems puzzling why participants chose smaller-sooner rewards at all in the short-delay tasks (SV and SN), given that subjects received the rewards only at the end of the experiment instead of after the delays. Could participants finish an experimental session earlier by choosing the SS instead of the LL options? This would point to possible opportunity costs associated with the LL reward option. Alternatively, could a demand effect contribute to the behavior? In addition, the authors report that participants might have experienced the "slot machine sound" that accompanied (virtual) reward deliveries as rewarding. This suggests that short-delay and long-delay tasks might not have been matched with regard to reward magnitude and modality, because in the LV task participants made choices between SS and LL monetary rewards, whereas in the SV and SN tasks they decided between sooner or later sounds that were experienced as secondary reinforcers (even though these had additionally also consequences for their payoffs).

Participants performed the three tasks in a fixed order (starting with the SN task, then performing the SV and NV tasks in alternating order). It seems possible that after the SN task participants internally set a decision criterion regarding the delay length/waiting time (relative to the given context) that they considered acceptable for a specific LL reward magnitude, which might be an alternative explanation for the significant correlations between the tasks. Due to the fixed task order, I see no way for empirically ruling out such an anchoring effect.

Following on from the previous point, I would assume that the implemented delays and magnitudes were distributed in the same fashion in the different tasks? If so, it may be less surprising that there are correlations between short and long-horizon tasks, at least when relative processing dominates (as in the primary experiment). Participants may simply transfer their choice patterns across similarly structured environments, possibly facilitated by participants desiring to be consistent in their proportions of choices. Conversely, if they were exposed to both short and long (hypothetical or real) delays within the same experiment, I would predict that their choice patterns look rather different.

Is the lack of correlation with BIS^-1^1 not an indication of limited validity of the task?

Statistical analysis:

The authors claim that their tasks show a high test-retest reliability, but it seems that the authors only tested for significant differences in discount parameters between the first and second half of blocks and between experimental sessions. To assess the reliability of a measure, however, it seems more appropriate to compute correlations, that is, to test whether the most impulsive subjects in the first experimental session stay the most impulsive ones in the other sessions.

Regarding the context effect observed in control experiment 2: first of all, I wonder whether such a context effect occurred also in the main experiment or in control experiment 1? The sample size is rather small in control experiment 2 (N=16), so it would be good to test the robustness of this effect in a larger sample size. Furthermore, in the main text, this effect is described as a "small but significant adaptation effect", which seems to contradict the headline of this section ("Strong effect of temporal context"). Given that this effect appears rather weak, whereas the discount factors for the SV and LV tasks are surprisingly similar, I think the authors should be more careful with rejecting the "costs of waiting"-hypothesis. I do not doubt the existence of such a context adaptation effect, but it might just be an additional factor besides the higher costs of waiting/opportunity costs in the SV relative to the LV task.

Minor Comments:

Please specify which learning stage the excluded participants did not pass.

*Reviewer #2:*

Authors examine patterns of delay discounting across three tasks, one non-verbal (experiential) and two verbal (one with long delay and one with short delay), and argue that individuals exhibit similar behavioral patterns scaled by the temporal context despite differences across conditions (verbal vs. non-verbal and short versus long delays).

The manuscript makes two valuable contributions. First, it provides evidence in support of comparing experiential studies in animal models to a large body of literature using verbal tests in humans. Second, it provides evidence that temporal context plays an important role in delay-discounting behavior.

I am generally positive, and have some specific comments that should help the authors improve the manuscript and make it more accessible.

Comments:

Results section, first paragraph: The description of reward for LV can be made clearer.

Results section first paragraph: The model description in the Results is too terse. I suggest explaining a little bit more the parameters since comparison of model parameters is an important part of Results. For example, "We found that k, for all tasks, had a log-normal distribution across our subjects," would not make a whole lot of sense to a non-expert.

Along the same lines, it would be good to start with a simulation of the model showing the expected effects of various key parameters. That would make understanding of the rest of the paper easier.

The explanation of model is clearer in Materials and methods but if the authors wish to make the results accessible to a larger readership, more details would help. Basically, I recommend unpacking the term "A 6 population level and 4 subject level parameters model." The subject level is explained well but the population level needs clarification. It appears that certain aspects of the package used for modeling are described as a black box.

Figure 2: It would help to discuss in Results the discrepancies between data and model. In some cases, there seem to be systematic errors that are ignored. Also, it would help to provide a measure of goodness of fit across subjects and tasks to get a better sense of how widespread such systematic errors were across subjects.

Figure 3, 4 and 5: Are the linear fits based on total least squares?

Paragraph three of subsection “Strong effect of temporal context”: HBA must be defined (it is defined in later use).

Overall, a very nice paper!

*Reviewer #3:*

This paper presents evidence that individual differences in temporal discounting have a degree of stability across large differences in both the scale of delays at stake and the format of the decision task. Human participants performed 3 different intertemporal choice tasks. The sample size (n=63) is on the modest side for an individual differences study, although there is quite a bit of data per individual. The tasks ranged from resembling standard rodent paradigms (with a nose-port-like interface, nonverbal cues, and directly experienced delays on the order of seconds) to resembling standard human paradigms (verbally cued delays on the order of days).

The findings add to our knowledge about the stability and generality of temporal discounting, and the comparability of human and animal experimental paradigms. This work is analogous and complementary to other recent research that compared risky decision making for probabilities derived from experience versus description.

I have a few comments and suggestions for potentially strengthening the manuscript.

1) As the authors aptly note (subsection “Subjects’ time-preferences are reliable across both verbal/experiential and second/day differences”), the correlation between two variables is independent of any difference in their means. There were a couple places where only one of these aspects of the data was quantified, and I thought it would be useful to see both:a) It would be useful to see test-retest reliability reported in terms of the correlation across sessions, in addition to the non-significant Wilcoxon signed-ranks test. This would more thoroughly support the statements about within-task reliability in the conclusions section (subsection “Stability of preferences”).b) For the days-vs-weeks control experiment, it would be nice to see the results of a paired-samples test comparing k-values in the two conditions, not just the correlation (subsection “Strong effect of temporal context” and Figure 5A). Although the correlation is high, the data in Figure 5A look like they might be systematically offset from the unity line.

2) It would be helpful to have more information about how the magnitude-delay pairs were constructed. For instance, what were the ranges of amounts and delays? Were they paired so as to cover a particular range of indifference k-values? (Subsection “Experimental Design” paragraph five and Figure 3—figure supplement 2 give partial information about this but not the complete picture.)

3) Subsection “Strong effect of temporal context” says k-values in the LV task were "almost equivalent (ignoring unexplained variance) to those in the SV task." I found this confusing because the preceding paragraphs emphasized that the LV k-values were significantly lower on average and also had higher variance than the SV k-values. Maybe this sentence just means to refer to the fact that LV and SV were correlated?

I also found it odd that the mismatch in units wasn't dealt with until this paragraph (i.e., whether k-value represents discounting per day or per second). I had assumed common units were being used when I first read the comparisons of k-values between tasks (paragraph five subsection “Subjects’ time-preferences are reliable across both verbal/experiential and second/day differences”). I think it would be helpful either to use matching units throughout, or point out (and explain) the choice not to at the outset.

4) A striking aspect of the results is the large difference in discount rates between short, directly experienced delays and long, non-experienced delays. In addition to considering the possibility that experienced delay is uniquely aversive (subsection “Cost of waiting vs. discounting future gains”), it would be interesting to consider the possible role of opportunity costs. I gather the NV and SV conditions didn't involve direct opportunity costs within the context of the experiment; that is, choosing longer delays didn't reduce the total number of trials, so the reward-maximizing strategy would always be to choose the larger reward? Did participants know in advance that they had a fixed number of trials (rather than having a time budget)? It would be interesting to know how participants' earnings compared to what they could have earned by following the reward-maximizing strategy. It would also be interesting to know whether they managed to finish the session and leave earlier than they would have by following the reward-maximizing strategy.

Minor Comments:

1) In the first paragraph of the Results section please say whether the given number of trials (160) is total or per task.

2) In the third paragraph of subsection “Stability of preferences” I would replace "eliminated" with "matched" or something similar. I initially read it as "ruled out," which is the opposite of the intended meaning.

3) Tables 2 and 3 would benefit from more descriptive legends. In particular, I initially misunderstood the Table 3 legend as meaning the outcome variable for this analysis was k-value variance (along the lines of the scaling effect mentioned for Table 4).

4) In Figure 5, I suggest noting explicitly in the legend that panels B-C pertain to the main experiment (and not the same experiment represented in panel A).

[Editors' note: further revisions were suggested prior to acceptance, as described below.]

Thank you for submitting your article "Time preferences are reliable across time-horizons and verbal vs. experiential tasks" for consideration by *eLife*. Your article has been reviewed by three peer reviewers, and the evaluation has been overseen by a Reviewing Editor and Timothy Behrens as the Senior Editor. The following individual involved in the review of your submission has agreed to reveal their identity: Mehrdad Jazayeri (Reviewer #2).

The manuscript has been improved but there are some remaining issues that we suggest you address before this is published.

1) The authors mention that boredom or opportunity costs may play a role with short delay. One may wonder whether these factors point at a potential difference between human and animal tasks as primary rewards could be ingested as they arrive. By extension, the presumed parallel of the present task with animal tasks may be smaller than assumed. This potential limitation could be mentioned in the discussion as it the comparison of human and animal research is a major motivation for the present study.

2) In the analysis in the final paragraph of subsection “Subjects’ time-preferences are reliable across both verbal/experiential and second/day differences”, which compares discount rates across tasks, it's now stated clearly that different units are used for k-values in the different tasks. But it might be beneficial to more fully describe the motivation for the analysis in light of this. Why is it of interest to test whether per-second discount rates in one task differ from per-day discount rates in another?

3) The Table 3 legend seems to have a typo (the 2nd occurrence of k_NV should be k_LV), and the abbreviation "Ev. Ratio" should be defined and explained (the evidence ratio is not introduced until paragraph seven of the “Analysis” section).

4) In subsection “Time and Reward Re-Scaling” I didn't understand why the k-values were referred to as "unit-free".

---

## [Author Response]

Reviewer #1:

Time preferences are investigated in both humans and animals, but temporal discounting tasks used in humans and animals differ with regard to duration of temporal delays and stimulus modalities. To empirically test the impact of these factors, the authors compared discount rates in humans between a novel non-verbal temporal discounting task with verbal tasks including short and long delays. The results reveal moderate to strong correlations between discount factors across tasks.There is much to like about the well-written manuscript: it addresses the important topic of the relevance of animal studies for understanding human behavior, the three temporal discounting tasks were rigorously matched with regard to choice structure, and several control analyses and control experiments were conducted to test alternative hypotheses. My main concern is that the current results allow no clear conclusions with regard to the theoretical question which mechanisms are shared (and which not) by the different temporal discounting tasks and that the parallels may be more imposed than it currently seems.

This is an important comment and in our revised draft we make clear where our study contributes to the literature. Specifically, in the Discussion section at we say: “this task can be applied to both human and animal research for direct comparison of cognitive and neural mechanisms underlying delay-discounting … animal models of delay-discounting may have more in common with short time-scale consumer behavior such as impulse purchases.… caution is warranted when reaching conclusions from the broader applicability of these models [short time-scale] to long-time horizon real-world decisions.”

Theoretical impact:The goal of the current experiments was to assess the impact of two methodological differences between human and animal studies, time horizon and stimulus modality (language-based vs. language-free). However, as the authors themselves write in the Discussion section, comparing discount rates across short and long time delays is not really novel, as this issue has already between addressed by previous studies.

This comment builds on the prior point that the earlier draft did not make explicit the contribution of this study. To the best of our knowledge and from undertaking a second review of the literature while completing the revision, our study offers several advantages relative to earlier work that compared decisions across time horizons. Our within-subject design removes the possibility that any differences arises due to unobserved confounds that differ between subjects. While, a few previous studies also considered within-subject variation either these studies had much smaller sample sizes and lower statistical power or used hypotheticals where none of the delayed rewards were actually experienced (e.g. Johnson et al., 2015). For those with low power, it is not a surprise that prior findings with small samples of within-subject variation did not yield conclusive results.

This leaves the comparison of different stimulus modalities as the main novel aspect of the current study, and unfortunately the results are not very conclusive here. This is because the correlations of the SN with the SV (0.54) and LV (0.36) are moderate in effect size, which suggests that the tasks measure both partially overlapping and at the same time also clearly dissociable constructs. While this is an interesting empirical finding in its own right, the implications of such moderate correlations for comparisons between human and animal studies are less clear.

We apologize if our descriptions of the correlations were unclear. We report correlations for main experiment in Table 1 and control experiments in Figure 4 and Figure 5. The Spearman correlation between non-verbal (NV) and short-verbal (SV) for the main experiment was 0.76 (Table 1), for Pearson correlation, 0.79. We have added the following text to the Discussion section on reliability "This correlation [0.79] is on par with test-retest reliability of personality traits (Viswesvaran and Ones, 2000)." The correlation between the short and long tasks was lower, and we acknowledge and discuss that difference at length in the discussion. We agree with the reviewer that the lower correlation between short and long tasks suggests that there are partially overlapping (but dissociable) mechanisms underlying short vs. long tasks. However, before our study the relative contribution of time-horizon vs. verbal-experiential to reliability of measured time-preferences was unknown.

The authors themselves appear to oscillate between different interpretations: for example, according to the manuscript title time preferences are reliable across verbal and experiential tasks, and also the Discussion section acknowledges that the three tasks share common cognitive and neural mechanisms, whereas at the same time "caution is warranted" when reaching conclusions from experiential short-delay paradigms in animals to verbal long-delay intertemporal decisions is humans. All this is true and at the same time somewhat trivial. From a theoretical perspective it would have been informative to get some idea about what these common and dissociable mechanisms might be. However the current study provides no answer to this question and further experiments would be needed.

We respectfully disagree with the reviewer. As to our "oscillatory" appearance, we have written the manuscript in a tone which we feel is honest to the data. The correlations between all three tasks were significant but they are not equal (Table 1). As to the "trivial" nature of our conclusions, there is no existing literature to provide guidance as to the relative reliability of time-preferences across the verbal/non-verbal gap. "To date, no published study with humans has examined discounting under a condition in which only symbolically presented information, and no specifically stated information, is provided about the delays and amounts." Vanderveldt et al., 2016.

Our main and 2nd control experiment (days vs. weeks) do provide evidence that the difference between waiting and postponing is what drives the differences across the time-horizons. We agree with the reviewer that future studies are required to fully understand the common and dissociable mechanisms between waiting and postponing. We are currently collecting brain imaging data to address that question to be published in a future manuscript.

Methodological concerns:The authors used the monetary rewards in the different temporal discounting tasks, but it seems puzzling why participants chose smaller-sooner rewards at all in the short-delay tasks (SV and SN), given that subjects received the rewards only at the end of the experiment instead of after the delays. Could participants finish an experimental session earlier by choosing the SS instead of the LL options? This would point to possible opportunity costs associated with the LL reward option.

We have addressed these questions in Figure 3—figure supplement 1 and in the Discussion: “…there could be opportunity costs related to how much subjects value their own time. We found that in the short tasks subjects with large discount factors also performed the task faster (Figure 3—figure supplement 1). If these subjects value their time more and thus have higher costs of waiting, then given our results Figure 3B there is a surprisingly large correlation between how much subjects value their time (in the short tasks) and how much they discount postponed rewards (in the long task)”.

Alternatively, could a demand effect contribute to the behavior?

A frequent critique of laboratory studies using human subject idea is the above idea of demand effect. One of the coauthors (Lehrer) has done some work with bargaining experiments where they created a design where one choice was strictly dominated by the other options (Embrey et al., 2014). Hence, if selected, it could reflect an experimenter demand effect. In their setting they found that it was only chosen less than 4% of the time and by very few subjects. In general, our sense from reading the broader literature in experimental economics is that the magnitude of demand effects is quite small.

Moreover, all of our tasks are computer-controlled and there is no interaction between the experimenter and the subject other than the instructions. Given that the vast majority of subjects' choices are sensitive to delays and rewards and are well-fit by a ~2 parameter model (per task) and that our estimates of these parameters are continually distributed over a range that is consistent with large scale studies of delay-discounting (Sanchez-Roige et al., 2018), we think that it is more likely that subjects are expressing their preferences than trying to adjust their behavior to a perceived belief in the purpose of the experiment.

We added our instructions for nonverbal and verbal experiments as Supplementary files to the manuscript for transparency. We read instructions out loud and answered questions of the participants. If questions about strategy arise, we left those questions not answered. In the nonverbal instructions there were no cues, except for “The only thing we can say now is USE THE MOUSE and react to different visual stimuli that appear on the screen and sound stimuli that you hear.” In the verbal experiments subjects are told about delays and waiting time. Both instructions are similar in describing the experiment: “play the game and earn coins”.

Embrey, M., Fréchette, G. R., and Lehrer, S. F. (2014). Bargaining and reputation: An experiment on bargaining in the presence of behavioural types. *The Review of Economic Studies, 82*(2), 608-631.

In addition, the authors report that participants might have experienced the "slot machine sound" that accompanied (virtual) reward deliveries as rewarding. This suggests that short-delay and long-delay tasks might not have been matched with regard to reward magnitude and modality, because in the LV task participants made choices between SS and LL monetary rewards, whereas in the SV and SN tasks they decided between sooner or later sounds that were experienced as secondary reinforcers (even though these had additionally also consequences for their payoffs).

This is an interesting point which we hadn’t considered. Inspired by the reviewer’s comment, we fit an expanded model to our data, which scales the reward differentially (per subject) in the short vs. long tasks. We found although subjects did perceive the two rewards as slightly different, the correlations between discount factors were slightly *increased*, and the temporal scaling between the short and long tasks was unchanged, thus supporting our original conclusions. See section “Controlling for differences in reward experience”.

Participants performed the three tasks in a fixed order (starting with the SN task, then performing the SV and NV tasks in alternating order). It seems possible that after the SN task participants internally set a decision criterion regarding the delay length/waiting time (relative to the given context) that they considered acceptable for a specific LL reward magnitude, which might be an alternative explanation for the significant correlations between the tasks. Due to the fixed task order, I see no way for empirically ruling out such an anchoring effect.

We thank the reviewer for bringing up the potential confounds of anchoring and reference point effects. First, we apologize if it wasn't clear, but subjects were put into two groups. All subjects did three sessions of NV first. These were followed by two sessions. Half of the subjects performed LV-SV-LV-SV in each session. Half of the subjects performed SV-LV-SV-LV in each session. To make this clearer, we have added this to the caption of Figure 1 "Note: The order of short and long delay verbal for sessions 4 and 5 was counter-balanced across subjects".

As such, the temporal proximity of SV and LV to NV differed between the groups. If our results were simply an anchoring effect, we would expect that the group to perform LV first would have a higher correlation to NV and the group performing SV first would have a higher correlation to NV. Instead we found:

SLSL NV vs SV r = 0.7662 p = 8.0880e-08

LSLS NV vs SV r = 0.8173 p = 1.1087e-07

SLSL NV vs LV r = 0.3742 p = 0.0268

LSLS NV vs LV r = 0.4313 p = 0.0219

Both types of correlations are slightly higher for LSLS than for SLSL. This is not consistent with an anchoring effect.

We have added an additional supplemental figure (Figure 3—figure supplement 2) to the paper to address these issues and added addition text.

“We further checked whether the correlations between discount factors in the three tasks may have arisen due to some undesirable features of our task design. For example, different subjects experienced the offers in different orders. Anchoring effects (Tversky and Kahneman, 1974; Furnham and Boo, 2011) may have set a reference point in the early part of the experiment that guided choices throughout the rest. As such, we repeated the analyses described in the previous paragraph, but we added 6 additional factors: the mean rewards, delays presented in the first block of the 2nd and 3rd non-verbal session and also the% of yellow choices made in those blocks. We reasoned that if anchoring effects were playing a role then subjects that were presented longer delays, or smaller rewards early in the experiment should have correlations between these factors and log(*𝑘𝑆𝑉*) or log(*𝑘𝑁𝑉)*. Likewise, if subjects were simply trying to be consistent with early choices, then the ‘% yellow’ in the early blocks would have an important influence. We tested the contribution of each factor by dropping it from the model to create a reduced nested model and using a likelihood ratio test against the full model. We found no evidence for anchoring effects or that subjects were simply trying to be consistent with their early choices.”

Following on from the previous point, I would assume that the implemented delays and magnitudes were distributed in the same fashion in the different tasks? If so, it may be less surprising that there are correlations between short and long-horizon tasks, at least when relative processing dominates (as in the primary experiment). Participants may simply transfer their choice patterns across similarly structured environments, possibly facilitated by participants desiring to be consistent in their proportions of choices. Conversely, if they were exposed to both short and long (hypothetical or real) delays within the same experiment, I would predict that their choice patterns look rather different.

We have added details to the Materials and methods about how the offers were presented. Although all subjects experienced all pairs of rewards and delays, the order was chosen randomly.

We included early choices in our regression mentioned above to minimize the potential that subjects were adopting a "consistency" strategy. However, even if subjects were driven by a desire to be internally consistent, that doesn't explain the wide distribution of time-preferences across subjects. Additionally, since the subjects first did the non-verbal (NV) task, it would be hard for them to be consistent (as a strategy) as they were themselves learning the mapping between sound features and rewards and delays.

Is the lack of correlation with BIS-11 not an indication of limited validity of the task?

We initially shared the concern of the reviewer, but after a closer read of the literature, we do not think that this is the case. The literature has mixed evidence for correlation between BIS and delay-discounting: some papers report significant positive correlations (Beck and Triplett, 2009; Mobini et al., 2007; Cosenza and Nigro, 2015), others don’t find significant correlations and suggest that delay discounting tasks might measure a different aspect of impulsivity (Mitchell, 1999; Fellows and Farah, 2005; Reynolds et al., 2006; Saville et al., 2010) We looked into components of BIS, since some researchers suggest that only some of them correlate with discounting coefficient (Fellows and Farah, 2005; Mobini et al., 2007; Beck and Triplett, 2009; Ahn et al., 2016). We found that correlation between NV log(k) and BIS nonplanning component is positive and significant at.05 level (r=0.3, p<0.05). We added addition text and analyses as follows:

“Prior research finds mixed evidence of the association between the BIS-11 and delay: some report significant positive correlations (Mobini et al., 2007; Beck and Triplett, 2009; Cosenza and Nigro, 2015), others don’t find significant correlations and suggest that delay discounting tasks might measure a different aspect of impulsivity (Mitchell, 1999; Fellows and Farah, 2005; Reynolds et al., 2006; Saville et al., 2010). Following earlier research that reports that components of the BIS score might drive the correlation with discounting (Fellows and Farah, 2005; Mobini et al., 2007; Beck and Triplett, 2009; Ahn et al., 2016), we next decomposed the score. Similar to others we found that correlation between BIS nonplanning component and log(𝑘𝑁𝑉) is positive and significant (Pearson𝑟=0.3,𝑝<0.05)."

Statistical analysis:The authors claim that their tasks show a high test-retest reliability, but it seems that the authors only tested for significant differences in discount parameters between the first and second half of blocks and between experimental sessions. To assess the reliability of a measure, however, it seems more appropriate to compute correlations, that is, to test whether the most impulsive subjects in the first experimental session stay the most impulsive ones in the other sessions.

We have computed correlations as recommended. We report these correlation in the paragraph starting in paragraph four of “Subjects’ time-preferences are reliable across both verbal/experiential and second/day differences”. “Consistent with existing research, we find that time-preferences are stable in the same task within subjects between the first half of the block and the second half of the block within sessions (time-preferences are measured as percent `yellow' choices, Wilcoxon signed-rank test, p = 0.35; Pearson r = 0.81, p < 10^-9) and also across experimental sessions that take place every two weeks: percent `yellow' choice between NV sessions (Wilcoxon signed-rank test, p = 0.47; Pearson r = 0.7, p < 10^-9), between SV sessions (Wilcoxon signed-rank test, p = 0.66; Pearson r = 0.82, p < 10^-9) and a slight difference between LV sessions (Wilcoxon signed-rank test, p < 0.1; Pearson r = 0.66, p < 10^-9) (Meier and Sprenger, 2015; Augenblick et al., 2015).

Regarding the context effect observed in control experiment 2: first of all, I wonder whether such a context effect occurred also in the main experiment or in control experiment 1? The sample size is rather small in control experiment 2 (N=16), so it would be good to test the robustness of this effect in a larger sample size. Furthermore, in the main text, this effect is described as a "small but significant adaptation effect", which seems to contradict the headline of this section ("Strong effect of temporal context"). Given that this effect appears rather weak, whereas the discount factors for the SV and LV tasks are surprisingly similar, I think the authors should be more careful with rejecting the "costs of waiting"-hypothesis. I do not doubt the existence of such a context adaptation effect, but it might just be an additional factor besides the higher costs of waiting/opportunity costs in the SV relative to the LV task.

We apologize for the confusion. The "Strong effect of temporal context" refers to the 5-order of magnitude shift in discount factors between short and long delays that we observed in the main experiment and also control experiment 1 (Figures 3, 4). The "small but significant" adaptation effect refers to the difference between the first few trials of a context and the rest of the trials in that context for the main results (n=63 subjects). We have moved this to its own figure (Figure 6) to make it clearer that it is not part of the days/weeks experiment.

We agree the cost of waiting plays a key role in the two short tasks. In fact, we believe that the reason that the correlation between the two short tasks is higher than the two verbal tasks is due to the importance of cost of waiting, as distinct from postponing reward. What we intended to communicate was that it seemed like too much of a coincidence that the shift in the mean discount factors between the short and long tasks was exactly the days to seconds shift.

Minor Comments:Please specify which learning stage the excluded participants did not pass.

Stage 2 of the learning stages, learning 'fixation', was the most difficult stage. This is reported in caption of Figure 1—figure supplement 1. We also added to the same caption that this is the stage subjects that were excluded couldn’t pass.

“Some subjects experienced difficulty with the learning to ‘fixate’ during learning stage 2. Subjects that didn’t pass learning stages stopped at this stage.”

In summary, we would like to thank this referee for a comprehensive review that alerted us to the possibility of alternative explanations for the observed patterns in the data. These comments led us to undertake additional robustness exercises, increasing our confidence in the main findings.

Reviewer #2:

Authors examine patterns of delay discounting across three tasks, one non-verbal (experiential) and two verbal (one with long delay and one with short delay), and argue that individuals exhibit similar behavioral patterns scaled by the temporal context despite differences across conditions (verbal vs. non-verbal and short versus long delays).The manuscript makes two valuable contributions. First, it provides evidence in support of comparing experiential studies in animal models to a large body of literature using verbal tests in humans. Second, it provides evidence that temporal context plays an important role in delay-discounting behavior.I am generally positive, and have some specific comments that should help the authors improve the manuscript and make it more accessible.Comments:Results section, first paragraph: The description of reward for LV can be made clearer.

We revised the description of reward for LV task by adding examples in the Results section:

“In the verbal long delay task, after each choice, subjects were given feedback confirming their choice (e.g. “Your choice: 8 coins in 30 days”) and then proceeded to the next trial. Unlike the short tasks, there was no sound of dropping coins nor visual display of coins. At the end of the session, a single long-verbal trial was selected randomly to determine the payment (e.g. a subject was notified that “Trial 10 from session 1 was randomly chosen to pay you. Your choice in that trial was 8 coins in 30 days”). If the selected trial corresponded to a subject having chosen the later option, she received her reward via an electronic transfer after the delay (e.g. in 30 days).”

The exchange rates from coins to money also differed between tasks. This is pointed out in the section “Strong effect of temporal context”

Results section first paragraph: The model description in the Results is too terse. I suggest explaining a little bit more the parameters since comparison of model parameters is an important part of Results. For example, "We found that k, for all tasks, had a log-normal distribution across our subjects," would not make a whole lot of sense to a non-expert.

We have made changes throughout the manuscript in order to make the results (and the model) easier to understand for a general audience.

For example, we provided additional explanation for model parameters:

“The subject level effects are drawn from a normal distribution with mean zero. In other words, the subject level effects reflect the difference of each subject relative to the mean across subjects. As such, the actual discount factor for the 𝑛𝑡h subject in the SV task, 𝑘𝑛,𝑆𝑉= 𝑒log(𝑘̂ 𝑆𝑉)+log(𝑘̇ 𝑛,𝑆𝑉) = 𝑘̂ 𝑆𝑉⋅𝑘̇ 𝑛,𝑆𝑉, where log(𝑘̂𝑆𝑉) represents the population level log discount factor for SV and log(𝑘̇ 𝑛,𝑆𝑉) represents the subjects level effect for subject 𝑛in SV. For the sake of brevity, we use the term ‘discount factor’ to refer to ‘log discount factor’ throughout the text. The population level parameters reflect the mean over all subjects. For example, if the mean discount factor across subjects was equal in all tasks, then the population level discount factor parameters would be also be equal. If all subjects were exactly twice as impulsive in short vs. long tasks, then that change would be reflected in the population level discount factor (𝑘𝑆𝑉= 2 ⋅𝑘𝐿𝑉→ log(𝑘𝑆𝑉) = log(𝑘𝐿𝑉) + log(2)), and the subject level parameters would be the same across tasks. If, on the other hand, impulsive subjects (relative to the mean) became more impulsive, and patient subjects became more patient, that would result in clear changes to subject level parameters, with relatively little change in the population level parameters (assuming the same scaling factor for impulsive and patient subjects).”

As well, we added Supplementary files that visually show the data and model fits and report (in text) the point estimates of the model parameters for each subject in all three experiments. We feel that these give an interested reader a very concrete connection between the data and the model parameters.

+ For main: Supplementary File 1

+ For 1st control exp: Supplementary File 2

+ For 2nd control exp: Supplementary File 3

Additionally, in order to help readers visualize how different discount factors lead to different subjective values of reward as a function of time we have added Figure 3—figure supplement 3.

Along the same lines, it would be good to start with a simulation of the model showing the expected effects of various key parameters. That would make understanding of the rest of the paper easier.

The expected effects of various key parameters are shown in the supplemental plot (Figure 2—figure supplement 4).

The explanation of model is clearer in Materials and methods but if the authors wish to make the results accessible to a larger readership, more details would help. Basically, I recommend unpacking the term "A 6 population level and 4 subject level parameters model." The subject level is explained well but the population level needs clarification. It appears that certain aspects of the package used for modeling are described as a black box.

The population level parameters are further described in subsection “Subjects’ time-preferences are reliable across both verbal/experiential and second/day differences”, cited above.

Figure 2: It would help to discuss in Results the discrepancies between data and model. In some cases, there seem to be systematic errors that are ignored. Also, it would help to provide a measure of goodness of fit across subjects and tasks to get a better sense of how widespread such systematic errors were across subjects.

We appreciate the reviewers comments. In the submitted version, we ignored these systematic errors because we found that our main results were robust to different methods of estimation. We apologize for that.

The difference between the model and data in Figure 2 are not very informative. We added the following text to the caption to explain "Note: The lines here are not a model fit to aggregate data, but rather reflect the mean model parameters for each group. As such, discrepancies between the model and data here are not diagnostic. See individual subject plots (Supplementary File 1) to visualize the quality of the model fits."

The subject plots provide a visual check on goodness of fit and also report the Bayesian r^2^ of the fit per subject-task. We found that for some subjects the Bayesian r^2^ was quite low. For NV, 9 subjects, for SV, 1 subject and for LV, 2 subjects had r^2^ < 0.2. We re-checked the Pearson correlations of log(k) excluding those subjects and did not find any change in our main effects:

NV vs. SV

r = 0.76 [0.62 0.85]

n = 54, p-value = 1.759e-11

SV vs. LV

r = 0.61 [0.42 0.74]

n = 60, p-value = 2.97e-07

NV vs. LV

r = 0.36 [0.10 0.58]

n = 52, p-value = 0.00841

The goodness of fits for each experiment and each task is depicted by plotting individual fits against real data for each subject of the main (Figure 2—figure supplement 2), control 1 (Figure 4—figure supplement 2) and control 2 (Figure 5—figure supplement 2) experiments and computing the Bayesian r^2^ for each subject-task (shown in figures).

Figure 3, 4 and 5: Are the linear fits based on total least squares?

We did not use total-least squares. That was an oversight, since the X-values are not under experimental control. We have re-generated the figures (and corresponding text) using total least square, x(y) and y(x) lines, since three lines are better than one (Huang et al., 2013).

For Figures 3AB, 4AB, 5A caption text added:”Three lines are vertical, horizontal and perpendicular (or total) least squares.”

Paragraph three of subsection “Strong effect of temporal context”: HBA must be defined (it is defined in later use).

Thank you for noticing that. We have now defined hierarchical Bayesian analysis model at its first appearance, but also decided to change the notation to a more common one:

“Subjects' impulsivity was estimated by fitting their choices with a Bayesian hierarchical model (BHM) of hyperbolic discounting with decision noise.”

Overall, a very nice paper!

Reviewer #3:

[…] I have a few comments and suggestions for potentially strengthening the manuscript.1) As the authors aptly note (subsection “Subjects’ time-preferences are reliable across both verbal/experiential and second/day differences”), the correlation between two variables is independent of any difference in their means. There were a couple places where only one of these aspects of the data was quantified, and I thought it would be useful to see both:a) It would be useful to see test-retest reliability reported in terms of the correlation across sessions, in addition to the non-significant Wilcoxon signed-ranks test. This would more thoroughly support the statements about within-task reliability in the conclusions section (subsection “Stability of preferences”).

We apologize for that oversight. We have now included those checks. “Consistent with existing research, we find that time-preferences are stable in the same task within subjects between the first half of the block and the second half of the block within sessions (time-preferences are measured as percent `yellow' choices, Wilcoxon signed-rank test, p = 0.3491; Pearson r = 0.81, p < 10^-9) and also across experimental sessions that take place every two weeks: percent `yellow' choice between NV sessions (Wilcoxon signed-rank test, p = 0.4721; Pearson r = 0.7, p < 10^-9), between SV sessions (Wilcoxon signed-rank test, p = 0.6613; Pearson r = 0.82, p < 10^-9) and a slight difference between LV sessions (Wilcoxon signed-rank test, p < 0.1; Pearson r = 0.66, p < 10^-16) (Meier and Sprenger, 2015; Augenblick et al., 2015)

b) For the days-vs-weeks control experiment, it would be nice to see the results of a paired-samples test comparing k-values in the two conditions, not just the correlation (subsection “Strong effect of temporal context” and Figure 5A). Although the correlation is high, the data in Figure 5A look like they might be systematically offset from the unity line.

Indeed, there is a systematic offset from the unity line for control experiment 2, so that the conversion rate from weeks to days is not 7 days = 1 week, we report that and we provide an additional supplemental figure (Figure 5—figure supplement 1) to discuss this effect.

“If subjects had ignored units then we would expect that log(𝑘𝑊) = log(𝑘𝐷) + log(7) = log(𝑘𝐷) + 1.95. Comparing the posteriors with that predicted shift, we can say that the shift is highly unlikely (𝑝 < 0.0001). Nonetheless, the discount factors in the two tasks were not equal. We observed a kind of amplification of preferences: the impulsive subjects were more impulsive in days than weeks and the patient subjects were more patient in days than weeks (Figure 5—figure supplement 1).”

2) It would be helpful to have more information about how the magnitude-delay pairs were constructed. For instance, what were the ranges of amounts and delays? Were they paired so as to cover a particular range of indifference k-values? (Subsection “Experimental Design” paragraph five and Figure 3—figure supplement 2 give partial information about this but not the complete picture.)

We are sorry for confusion, in the updated manuscript we write:

“Across trials, the delay and the magnitude of the sooner option were fixed (4 coins, immediately), later options were drawn from all possible pairs of five magnitudes and delays (25 different offers, Materials and methods).”

“There were 25 different “later” options presented in each task: all possible combinations of 5 delays (3, 6.5, 14, 30, 64) and 5 reward magnitudes (1, 2, 5, 8, 10).”

3) Subsection “Strong effect of temporal context” says k-values in the LV task were "almost equivalent (ignoring unexplained variance) to those in the SV task." I found this confusing because the preceding paragraphs emphasized that the LV k-values were significantly lower on average and also had higher variance than the SV k-values. Maybe this sentence just means to refer to the fact that LV and SV were correlated?

We apologize for the confusion: yes we intended to convey the message that they were surprisingly close given the differences between seconds and days. We have edited that text as follows:

“But, we found that the discount factors in the LV task, *𝑘𝐿𝑉*, were close to those in the other tasks (within ~1 log-unit) (Figure 3C).”

I also found it odd that the mismatch in units wasn't dealt with until this paragraph (i.e., whether k-value represents discounting per day or per second). I had assumed common units were being used when I first read the comparisons of k-values between tasks (paragraph five subsection “Subjects’ time-preferences are reliable across both verbal/experiential and second/day differences”). I think it would be helpful either to use matching units throughout, or point out (and explain) the choice not to at the outset.

We thank you for your comment and tried to clarify the use of units throughout the paper. The reason for fitting the units of experiment at first, rather than converting to common units was due to not having an *a priori* hypothesis about the relationship between the tasks. Thus, we first checked rank correlations between discount factors in different tasks. After seeing the results, we decided that utilizing the units of the task makes it easier to see that the scaling between short and long tasks almost matches the units conversion. We now mention the units in many places throughout the manuscript. It appears first in the text on reported in paragraph two of subsection “Subjects’ time-preferences are reliable across both verbal/experiential and second/day differences”:

“Since we did not *ex ante* have a strong hypothesis about how the subjects' impulsivity measures in one task would translate across tasks, we fit subjects choices in the units of the task (i.e. seconds and days), examined ranks of impulsivity at first and found significant correlations across experimental tasks (Table 1).”

Units are also indicated on Figure 3A,B and in the caption.

4) A striking aspect of the results is the large difference in discount rates between short, directly experienced delays and long, non-experienced delays. In addition to considering the possibility that experienced delay is uniquely aversive (subsection “Cost of waiting vs. discounting future gains”), it would be interesting to consider the possible role of opportunity costs. I gather the NV and SV conditions didn't involve direct opportunity costs within the context of the experiment; that is, choosing longer delays didn't reduce the total number of trials, so the reward-maximizing strategy would always be to choose the larger reward? Did participants know in advance that they had a fixed number of trials (rather than having a time budget)? It would be interesting to know how participants' earnings compared to what they could have earned by following the reward-maximizing strategy. It would also be interesting to know whether they managed to finish the session and leave earlier than they would have by following the reward-maximizing strategy.

We thank the reviewer for this feedback. We agree that the difference in discount rates between short and long tasks is quite striking! We have reviewed additional literature on opportunity costs added some relevant references to the Discussion: "Second, it may be that with short delay tasks we are capturing cost of waiting while long delay tasks measure delay-discounting. The costs of waiting could take several forms (Paglieri, 2013). One form is the cost of boredom (Mills and Christoff, 2018). Subjects could find it painful to sit and wait, starting at the clock, during the delay. Additionally, there could be opportunity costs related to how much subjects value their own time. We found that in the short tasks subjects with large discount factors also performed the task faster (Figure 3—figure supplement 1). If these subjects value their time more and thus have higher costs of waiting, then given our results Figure 3B there is a surprisingly large correlation between how much subjects value their time (in the short tasks) and how much they discount postponed rewards (in the long task). Regardless of the precise form of the costs of waiting (Chapman, 2001; Paglieri, 2013; Navarick, 2004) in order for these costs to explain the temporal scaling we observed between short and long tasks, relative to the costs of postponing, they would have to be, coincidentally, close in value to the number of seconds in a day.”

As to your specific questions: subjects experiences a fixed number of trials but were not told so. Choosing longer delay did not reduce the total number of trials. We have added the subject instructions as an appendix to make this clear to readers. The only info regarding this matter that was provided was in the consent form, it said “the session won’t take longer than 1 hour”.

We looked into fraction of total rewards and total time spent by subjects for each task (Figure 3—figure supplement 1). As expected both were correlated with discount factor. Plus, subjects that took longer times to finish the session were also slower (mouse clicking, etc.) in non waiting time as shown in the supplemental figure.

Minor Comments:1) In the first paragraph of the Results section please say whether the given number of trials (160) is total or per task.

Thank you for this comment. We reported in the Results section that the given number of trials is per task:

“Across sessions, at least 160 trials in each task were conducted after learning (Materials and Methods, Figure 1—figure supplement 1).”

We also report the total number of trials in the main experiment.

2) In the third paragraph of subsection “Stability of preferences” I would replace "eliminated" with "matched" or something similar. I initially read it as "ruled out," which is the opposite of the intended meaning.

We see how that could be confusing. We have changed the text to "Our control study using delays of days vs. weeks compared tasks with different scales but did not differ in the experience of the delayed rewards, as in LV, only (at most) one delayed reward was experienced for both days and weeks tasks."

3) Tables 2 and 3 would benefit from more descriptive legends. In particular, I initially misunderstood the Table 3 legend as meaning the outcome variable for this analysis was k-value variance (along the lines of the scaling effect mentioned for Table 4).

We updated Table 3 (now Table 2) legend for clarity:

“Relative contributions of two gaps to variance in log(k) (2-factor model comparison with two reduced 1-factor models)”

4) In Figure 5, I suggest noting explicitly in the legend that panels B-C pertain to the main experiment (and not the same experiment represented in panel A).

We realized that was confusing. We decided to split this Figure into two: Figure 5 now has only control experiment 2 results, whereas Figure 6 only displays the adaptation effect.

Figure 6 caption reads: “(A,B) Main experiment early trials adaptation effect.”

[Editors' note: further revisions were suggested prior to acceptance, as described below.]

The manuscript has been improved but there are some remaining issues that we suggest you address before this is published.1) The authors mention that boredom or opportunity costs may play a role with short delay. One may wonder whether these factors point at a potential difference between human and animal tasks as primary rewards could be ingested as they arrive. By extension, the presumed parallel of the present task with animal tasks may be smaller than assumed. This potential limitation could be mentioned in the discussion as it the comparison of human and animal research is a major motivation for the present study.

Animals certainly also experience opportunity costs and there is substantial evidence that they experience boredom as well. There is interesting work on the evolutionary pressures that selected for time-discounting, but we feel that this is outside the scope of our discussion. We have added the following lines to the Discussion:

In the first paragraph of “Subjective Scaling of Time”

Whether or not the secondary reinforcer used in our task is experienced in an analogous way to primary reinforcers used in animal studies may limit the degree of overlap in underlying neural mechanisms.

In the third paragraph:

One form is the cost of boredom (*Mills and Christoff, 2018*); a feeling which animals may also experience (*Wemelsfelder, 1984*)

2) In the analysis in the final paragraph of subsection “Subjects’ time-preferences are reliable across both verbal/experiential and second/day differences”, which compares discount rates across tasks, it's now stated clearly that different units are used for k-values in the different tasks. But it might be beneficial to more fully describe the motivation for the analysis in light of this. Why is it of interest to test whether per-second discount rates in one task differ from per-day discount rates in another?

Re-reading that section, we can appreciate the reviewers’ comment. We now include this in that section:

“Note, that *𝑘𝑁𝑉* and *𝑘𝑆𝑉* have units of Hz (1∕*𝑠*),but *𝑘𝐿𝑉* has units of 1∕*𝑑𝑎𝑦*. Thus, while the 95% credible intervals of the means of log(*𝑘*) are overlapping for the three tasks when expressed in the units of each task, the mean log(*𝑘𝐿𝑉*) is in fact shifted to -14.86 when *𝑘𝐿𝑉* is expressed in units of 1/s. We further analyze and discuss this scaling subsequently, but first we compare log(*𝑘*) in the units of each task, in consideration of subjects potentially ignoring the time units in their choices (Furlong and Opfer, 2009; Cox and Kable, 2014). We find that, on average, subjects were most patient in LV, then SV then NV *Table 3*). Note, that a shift of 1 log-unit is substantial. For example, a subject with log(*𝑘𝑆𝑉*) ≈ −3 would value 10 coins at half its value in just 20 seconds. But for log(*𝑘𝑆𝑉*) ≈ −4 the coins would lose half their value in 55 seconds (Figure 3—figure supplement 3).

And in the caption for Table 3 we added:

“Expressing log(*𝑘𝐿𝑉*) in units of 1/s (for direct comparison with the other tasks) results in a negative shift in log(*𝑘𝐿𝑉*) and even larger differences in means without changing the difference between standard deviations.

3) The Table 3 legend seems to have a typo (the 2nd occurrence of k_NV should be k_LV), and the abbreviation "Ev. Ratio" should be defined and explained (the evidence ratio is not introduced until paragraph seven of the “Analysis” section).

Thank you for pointing that out. The typo is fixed and evidence ratio is explained in the caption.

4) In subsection “Time and Reward Re-Scaling” I didn't understand why the k-values were referred to as "unit-free".

We apologize for the confusion. What we meant was that we originally compared K in units of the respective task. “Unit-free” was a poor word choice and has been removed from the two places it appeared in the paper.